# Addressing Biases in Arctic-Boreal Carbon Cycling in the Community Land Model Version 5

Leah Birch[1], Christopher R. Schwalm[1], Sue Natali[1], Danica Lombardozzi [2], Gretchen Keppel-Aleks[3], Jennifer Watts[1], Xin Lin[3], Donatella Zona[4], Walter Oechel[4], Torsten Sachs[5], Thomas Andrew Black[6], and Brendan M. Rogers[1]

[1]Woodwell Climate Research Center, Falmouth, MA
[2]National Center for Atmospheric Research, Boulder, CO
[3]University of Michigan, Ann Arbor, MI
[4]San Diego State University, San Diego, CA
[5]GFZ German Research Centre for Geosciences, Potsdam, Germany
[6]University of BC, Vancouver, BC, Canada

**Correspondence:** Leah Birch (lbirch@woodwellclimate.org) and Brendan Rogers (brogers@woodwellclimate.org)

**Abstract.** The Arctic-boreal zone (ABZ) is experiencing amplified warming, actively changing biogeochemical cycling of vegetation and soils. The land-to-atmosphere fluxes of $CO_2$ in the ABZ have the potential to increase in magnitude and feedback to the climate causing additional large scale warming. The ability to model and predict this vulnerability is critical to preparation for a warming world, but Earth system models have biases that may hinder understanding the rapidly changing ABZ carbon fluxes. Here we investigate circumpolar carbon cycling represented by the Community Land Model 5 (CLM5.0) with a focus on seasonal gross primary productivity (GPP) in plant functional types (PFTs). We benchmark model results using data from satellite remote sensing products and eddy covariance towers. We find consistent biases in CLM5.0 relative to observational constraints: (1) the onset of deciduous plant productivity to be late, (2) the offset of productivity to lag and remain abnormally high for all PFTs in fall, (3) a high bias of grass, shrub, and needleleaf evergreen tree productivity, and (4) an underestimation of productivity of deciduous trees. Based on these biases, we focus model development of alternate phenology, photosynthesis schemes, and carbon allocation parameters at eddy covariance tower sites. Although our improvements are focused on productivity, our final Model Recommendation results in other component $CO_2$ fluxes, e.g. Net Ecosystem Exchange (NEE) and Terrestrial Ecosystem Respiration (TER), that are more consistent with observations. Results suggest that algorithms developed for lower latitudes and more temperate environments can be inaccurate when extrapolated to the ABZ, and that many land surface models may not accurately represent carbon cycling and its recent rapid changes in high latitude ecosystems, especially when analyzed by individual PFTs.

## 1 Introduction

As the atmospheric concentration of $CO_2$ continues to rise, the Arctic-boreal Zone (ABZ) is expected to continue to warm more rapidly than the rest of the globe (Serreze and Francis, 2006; Serreze and Barry, 2011). The impacts of this accelerated warming are manifest across all major components of the ABZ — the cryosphere, hydrosphere, and biosphere (Duncan et al.,

2020). The multifaceted ABZ response to warming includes accelerated carbon cycling (Jeong et al., 2018), permafrost thaw, intensification of disturbance regimes (Alexander and Mack, 2016), changes in snow cover and ecosystem water availability (Callaghan et al., 2011; Biancamaria et al., 2011), and shifts in vegetation structure and composition (Beck et al., 2011; Forkel et al., 2016; Searle and Chen, 2017). These changes in the whole ABZ terrestrial ecosystem structure and function have important implications for global climate, given the region's strong biophysical coupling (Bonan et al., 1992; Bala et al., 2007; Rogers et al., 2013, 2015) and large, and potentially vulnerable, reservoirs of below and aboveground carbon, especially in the permafrost zone (Shaver et al., 1992; McGuire et al., 2009, 2010; Koven et al., 2015; Parazoo et al., 2018b; Natali et al., 2019; McGuire et al., 2018).

The responses of carbon cycling in the ABZ to changes in global climate are complex, interconnected, and may have compensating effects (Welp et al., 2016). For example, air and soil warming, in conjunction with a lengthening of the annual non-frozen period across the ABZ (Kim et al., 2012), stimulate plant productivity directly and indirectly through increased nutrient and water availability (Natali et al., 2014; Salmon et al., 2016). Warming and $CO_2$ fertilization have contributed to widespread "greening" across the ABZ, including shrubification (Myers-Smith et al., 2011, 2015) and northward treeline expansion (Lloyd and Fastie, 2003; Chapin et al., 2005), i.e. the encroachment of trees and shrubs into tundra regions. However, rapid warming across much of the ABZ is also accelerating decomposition, causing drought stress in warmer and drier landscapes (Carroll et al., 2011; Walker and Johnstone, 2014; Walker et al., 2015; Carroll and Loboda, 2017), and intensifying disturbance regimes such as wildfire and insect outbreaks (Turetsky et al., 2011; Kasischke et al., 2010; Rogers et al., 2018; Hanes et al., 2019); all of which contribute to the increasingly observed patterns of "browning" in the ABZ (Verbyla, 2011; Elmendorf et al., 2012; Phoenix and Bjerke, 2016).

As an emergent property of global change drivers in the ABZ, the seasonal cycle of $CO_2$ exchange across the ABZ has been experiencing changes in timing and magnitude of fluxes. Most critically regarding the magnitude of carbon fluxes, the atmospheric $CO_2$ concentration in the ABZ has been measured to be increasing between 30-60% during the last 60 years (Keeling et al., 1996; Randerson et al., 1999; Graven et al., 2013; Liptak et al., 2017; Jeong et al., 2018). Our current knowledge of the ABZ seasonal cycle of $CO_2$ suggests that much of the observed change in seasonal amplitude is due to increased vegetation productivity during the growing season, a result of $CO_2$ fertilization and warming (Forkel et al., 2016; Ito et al., 2016; Zhao et al., 2016). At the same time, fall and winter respiration constitute a large portion of the annual $CO_2$ budget (Euskirchen et al., 2014; Natali et al., 2019) and have been increasing with climate change (Belshe et al., 2012; Piao et al., 2008), making the implications for net sink-source dynamics uncertain (Ciais et al., 1995; McGuire et al., 2018). Hence, the current and anticipated state of carbon source-sink dynamics remains an open question in part due to the uncertainty in the dominant mechanisms and differential responses governing carbon fluxes across the ABZ.

Ground observations, satellite products, and process-based climate models are all used to understand interactions and feedbacks between changing environmental conditions and carbon cycling in the ABZ. *In situ* observations of carbon fluxes are required for mechanistic understanding but are often limited across time and space, especially in large and remote regions with extreme temperatures, like the ABZ (Virkkala et al., 2018, 2019). For example, respiration during the winter has long been assumed to be effectively zero, but better technology has slowly allowed the seasonal cycle story to grow (Natali et al., 2019).

Satellite observations provide near complete coverage in space and time, but are indirect observations of ecosystem properties, are challenging in the ABZ due to low insolation in the winter months and extreme snow storms, and contain a variety of uncertainties related to sensor properties, atmospheric contamination, and processing (Duncan et al., 2020). The brevity of the growing season and lack of light in the ABZ throughout the year also contributes to biases in satellite measurements (Randerson et al., 1997). Process-based models, or terrestrial biosphere models (TBMs), are a particularly invaluable resource for examining mechanisms across spatial and temporal scales, even projecting carbon cycle feedbacks in the future under varying socioeconomic scenarios (Taylor et al., 2012; Eyring et al., 2016, CMIP5 and 6). However, due to different formulations, assumptions, mechanisms, model inputs, and parameterizations, TBMs display a wide range of $CO_2$ source-sink dynamics in the ABZ (Fisher et al., 2014; Huntzinger et al., 2013) and biases compared to observations (Schwalm et al., 2010; Schaefer et al., 2012). Given the criticality of the ABZ to future global carbon balance and the heterogeneity of landscape responses to warming, it is a high priority to understand and address the current biases in TBM carbon cycling.

The Community Land Model version 5.0 (CLM5.0) is the land component of the Community Earth System Mode version 2.0 (CESM2.0). CLM is one of the most widely-used land surface models and contributes to many global intercomparisons (Zhao and Zeng, 2014; Peng et al., 2015; Ito et al., 2016) and future climate projections relevant for scientists and policymakers (Piao et al., 2013, eg, IPCC). The current state-of-the-art release of the Community Land Model (Lawrence et al., 2019, CLM) incorporates several improvements to climatic fluxes and biogeochemistry relevant for the ABZ. A general improvement was observed globally for CLM5.0 compared to past versions of the model (i.e. CLM 4.0 and 4.5). However, a high bias in photosynthesis or gross primary productivity (GPP) at high latitudes remains a well-documented issue (Wieder et al., 2019) in CLM5.0. Thus, we explore the simulation of GPP along with the net ecosystem exchange (NEE) and terrestrial ecosystem respiration (TER) in order to identify biases in simulation of the seasonal carbon balance.

This study assesses the ability of CLM5.0 to accurately represent $CO_2$ fluxes with gridded model simulations, identifies deficiencies in the simulation of ABZ carbon fluxes, and provides a Model Recommendation for application in the ABZ. We provide a step-by-step diagnosis of the major factors contributing to biases in the simulation of the seasonal cycle of $CO_2$ fluxes in CLM5.0. We use FLUXCOM (Jung et al., 2017, 2019, a gridded product based on machine learning), the International Land Model Benchmarking Project (Collier et al., 2018, ILAMB) to assess model results, and in situ data from FluxNet (https://fluxnet.fluxdata.org) and Ameriflux (https://ameriflux.lbl.gov/). We focus our development on the simulation of $CO_2$ fluxes for each ABZ vegetation type in CLM5.0, representing the tundra and boreal forest. We use point-based simulations at eddy covariance (i.e., EC or flux) tower sites to inform the failure or success of each model development test of the phenology and photosynthesis modules in CLM5.0. We validate model development using gridded products and additional flux towers (withheld from the initial model development) before making our final Model Recommendation. As a result, we identify and resolve many of the known biases in the representation of phenology (Richardson et al., 2012), photosynthesis (Lawrence et al., 2019), and carbon allocation in CLM5.0, allowing a more realistic representation of carbon cycling in this rapidly changing ecosystem.

## 2 Methods

We investigate the seasonal cycle of ABZ $CO_2$ fluxes with CLM5.0 due to its widespread use and significant model improvements from the previous version. These updated processes include snow physics related to snow age and density, canopy snow interactions, active layer depth, groundwater movement, soil hydrology and biogeochemistry, and river transport (Li et al., 2013). Moving away from globally-constant values of plant traits that are challenging to measure, carbon and nitrogen cycle representations now use prognostic leaf photosynthesis traits (Ali et al., 2016, the maximum rate of electron transport or $J_{max}$ and the maximum rate of carboxylation or $V_{cmax}$), carbon costs for nitrogen uptake, leaf nitrogen optimization, and flexible leaf stoichiometry. Stomatal physiology were updated with the Medlyn conductance model, replacing the Ball-Berry (Medlyn et al., 2011). Additionally plant hydraulics have undergone recent improvement in more realistic stress representation (Kennedy et al., 2019). One primary goal with these improvements was to allow for more physically-based parameters that could be informed by observational ecological data, ultimately allowing for better fidelity with hydrological and ecological processes. Land cover inputs to CLM5.0 were updated to capture transient land use changes from the satellite record.

### 2.1 Pan-Arctic CLM5.0 Simulation

We run CLM5.0 at 0.5° by 0.5° grid resolution with meteorology inputs (rainfall, snowfall, 2m air temperature, 2m specific humidity, surface pressure, downward shortwave radiation, downward longwave radiation, 10m wind speed, and cloud cover fraction) from the Global Soil Wetness Project (GSWP3v1,http://hydro.iis.u-tokyo.ac.jp/GSWP3/ ), which is a standard forcing dataset in the Land Surface, Snow and Soil Moisture Model Intercomparison Project (Van den Hurk et al., 2016, LS3MIP). GSWP3v1 has been shown to be appropriate and the least biased forcing dataset for CLM5.0 simulations in the ABZ (Lawrence et al., 2019). We begin a simulation of the CLM5.0 release in 1850 and run through 2014 including default time series inputs of $CO_2$, aerosol deposition, nitrogen deposition, and land use change (Lamarque et al., 2010; Lawrence et al., 2016), which are available on NCAR's Cheyenne system (Computational and Laboratory, 2017). We implement a regional simulation of CLM5.0 north of 40° N across both hemispheres, allowing us to focus exclusively on ABZ processes. We confirm the improvements made to the newly updated CLM version 5.0 (Lawrence et al., 2019) through a comparison of CLM version 4.5 with the same input datasets.

For our control CLM5.0 simulation, we use an available equilibrated 1850 intialization on NCAR's Cheyenne system (Computational and Laboratory, 2017) with spun-up carbon pools. After model development, we again spin the model using this initial dataset, and we find GPP to equilibrate quickly, within 20 years (Supplement Fig. S1). To be conservative, we spin-up the model with our recommended model development versions for 100 years to ensure carbon fluxes have come to equilibrium. Then we use this equilibrated state as initial conditions in a production run simulation beginning in 1850 with all the same configuration and climatology as the CLM5.0 release control simulation.

Typical of land surface models, CLM5.0 represents vegetation through broad plant functional types (PFTs). CLM5.0 represents ABZ vegetation using five PFTs: needleleaf evergreen boreal trees (NETs), needleleaf deciduous boreal trees (NDTs), broadleaf deciduous boreal trees (BDTs), deciduous boreal shrubs (hereafter "shrubs"), and arctic C3 grasses (hereafter

"grasses"). We focus model development on PFT-specific comparisons, which allows a direct comparison with observational data. Any improvements to PFT specific carbon flux simulations have implications for changing vegetation distributions in the ABZ.

## 2.2 Model Benchmarking and Validation with FLUXCOM and ILAMB

Benchmarking is the process of quantifying model performance based on observational data considered to be the expected value or truth. We use FLUXCOM (Tramontana et al., 2016; Jung et al., 2017, 2019) to benchmark gridded $CO_2$ fluxes (i.e., gross primary productivity, terrestrial ecosystem respiration, and net ecosystem exchange, or GPP, TER, and NEE) in CLM5.0. FLUXCOM is an upscaled machine learning product based on FLUXNET eddy covariance towers. As a global product, FLUXCOM is particularly useful for filling spatial gaps in tower observations, especially in the relatively data-sparse ABZ. This product uses multiple reanalysis datasets and machine learning methods to train multiple predictors at flux tower sites. The resulting product is the mean of those ensembles, which also allows standard error to be calculated. We use the standard deviation around the mean to identify successful model development. Machine learning is a useful tool for this type of gap filling, as it does not care about geoghraphic locations, just the predictor space, which are the fluxes and environmental conditions. FLUXCOM is unable to simulate fluxes from fires and $CO_2$ fertilization accurately, which is why we focus model development on averages over the past couple decades, rather than specific years with forest fires and the increasing $CO_2$ amplitude trend. Any systemic problems with FLUXNET data would also exist within FLUXCOM, but validations of regions with sun-induced chlorophyll fluorescence (Köhler et al., 2015, SIF) add confidence in the FLUXCOM product. Additionally, derived from global MODIS-based vegetation layers Sulla-Menashe and Friedl (2019), FLUXCOM has the ability to generate PFT-specific output. Although there are inconsistencies between the PFT classifications, such as the representation of "mixed forests" in FLUXCOM, this allows direct comparisons of the PFT-specific fluxes represented by CLM5.0.

For an independent set of comparisons that includes additional environmental variables, we also use the International Land Model Benchmarking System (Collier et al., 2018, ILAMB). ILAMB is an open-source land model evaluation system that provides a uniform approach to benchmarking and scoring model fidelity. It is a powerful tool to quickly and thoroughly investigate biases, seasonality, spatial distribution, and interannual variability in climate model output. We use ILAMB to benchmark fluxes of $CO_2$, moisture, and heat, in addition to several land surface properties essential for climate responses and feedbacks such as albedo and leaf area index (LAI). Although the focus of our model development is on GPP, TER and NEE tend to respond strongly to changes in productivity (Chapin et al., 2006; Schaefer et al., 2012; Chen et al., 2015). We benchmark these additional interdependent properties in ILAMB to ensure our development generates systematic improvements.

## 2.3 Point Simulation Protocol

Although FLUXCOM is an invaluable tool to fill spatial and temporal gaps in tower observations across the ABZ, it is by definition not as accurate as direct *in situ* observations of $CO_2$ fluxes, for instance measurements from EC towers, which also include helpful ancillary information such as detailed vegetation composition. After benchmarking the aggregated grid cell fluxes, we assess model performance at specific EC towers that measure year-round seasonal $CO_2$ fluxes, which is a standard

model development procedure (Stöckli et al., 2008a). We aggregate fluxes of $CO_2$ to monthly means from flux towers in the ABZ that are part of the FluxNet (https://fluxnet.fluxdata.org) and Ameriflux (https://ameriflux.lbl.gov/) networks. We screen the tower records to determine if the PFT type in CLM5.0 corresponds to the vegetation described by tower metadata. We choose towers and grid cells with at least three years of sample data before 2014, as that is the end data of GSWP forcing data for CLM5.0. Collectively, the chosen towers that conform to our data requirements span all PFT types over the ABZ (Supplement Table S1). We divide our observational data into model development sites vs. evaluation sites to prevent overfitting of parameters. Our chosen model development sites are US-EML (Belshe et al., 2012), CA-QC2 (Margolis, 2018), CA-OAS (Black, 2016, BDT), and RU-SKP (Maximov, 2016), which encompass all of the CLM5.0 PFTs. We verify our work using additional flux tower measurements from FI-SOD (Aurela et al., 2016), RU-Tks (Aurela, 2016), CA-Sf1 (Amiro, 2016), US-Atq (Oechel et al., 2014), RU-Sam (Kutzbach et al., 2002-2014; Holl et al., 2019; Runkle et al., 2013), and CA-Gro (McCaughey, 2016).

During our model development process, we examine phenology and photosynthesis schemes in CLM5.0 by running point simulations starting in 1901 using the same inputs as our gridded simulations. For each model issue listed in Section 2.4, we iteratively test hypotheses and ranges of parameters values that may improve the simulation at the representative towers. Points simulations allow for rapid deployment of model tests, while also conserving compute resources. This speed of computation is invaluable for our multiple model development trajectories. We find that for our focus on phenology and photosynthesis, carbon fluxes equilibrate rapidly and a 20-year spin up is sufficient for point simulations (Supplement Fig. S1). We run the point simulations through 2014 and compared the years measured by flux towers with the same years simulated by CLM5.0. We acknowledge that the climatology experienced by a given flux tower and the reanalysis data used as a model input the model are different. Thus, we focus on the mean seasonal behavior of the flux towers and CLM5.0 to guide model development. The yearly variance serves as an uncertainty range for our characterization of flux tower behavior. Additionally, using the mean monthly $CO_2$ fluxes as calibration data can prevent over-fitting of CLM5.0 parameters. Each model development simulation for a specific PFT is also run for the other PFTs at the development sites (CA-QC2, CA-OAS, US-EML, and RU-SKP). After finalizing a given model development scheme, we implement the updates at the withheld EC sites (Supplement Table S1) and then in a gridded fashion across our ABZ regional domain.

## 2.4 Model Development

We identify several issues in the phenology and photosynthesis schemes in CLM5.0 for the ABZ, which are detailed in Section 3.1. These can be categorized by (i) extrapolation of schemes and parameterizations designed for temperate vegetation, (ii) biases in the prediction of leaf photosynthetic traits and (iii) mis-specified carbon allocation parameters. We also incorporate a bug fix as detailed in the Supplement Section 3.

### 2.4.1 Phenology Onset

The representation of spring and autumn phenology for deciduous trees, shrubs and grasses in CLM5.0 is based on a study of the conterminous United States (White et al., 1997) and extended to the ABZ. In the extratropics, plants initiate their photosynthetic

growing season in response to various climatic factors in spring, reach peak productivity in summer, and enter dormancy in autumn. This is parameterized in CLM5.0 by allowing spring onset to begin once a threshold for cumulative growing degree
days is met, as determined by White et al. (1997) using relationships between temperatures and the satellite based Normalized Difference Vegetation Index (NDVI). Thus, in CLM5.0, onset is based on relationships derived from temperate latitudes and extrapolated to the ABZ. We find that this parameterization requires relatively warm temperatures for the ABZ before onset can begin, which causes a delay in the beginning of the growing season for deciduous plants in the ABZ.

To implement a more mechanistic approach to onset in the ABZ, we identify environmental thresholds that correspond
to physiological changes during spring onset in high latitudes. Field observations consistently demonstrate that productivity initiation in the ABZ is governed by the cessation of freezing temperatures (Ueyama et al., 2013; Stöckli et al., 2008b) and the availability of soil water(Goulden et al., 1998). Additionally snow cover has been shown to influence the start of the growing season (Høye et al., 2007; Semenchuk et al., 2016). We use daily output from FLUXCOM and flux towers to identify the initiation of GPP in spring for different PFTs. We then compare the timing of productivity to a variety of CLM5.0 environmental
variables known to correspond strongly with GPP onset (Chapin III and Shaver, 1996; Starr and Oberbauer, 2003; Borner et al., 2008), including soil temperature, soil moisture, soil ice content, air temperature, liquid and ice precipitation, snow depth, and latent and sensible heat fluxes (Supplement Fig. S2). We find that soil temperature (and thus soil ice content in the third soil layer with ~10 cm depth), minimum 2m temperature, and snow cover undergo notable state transitions around the timing of GPP onset, enabling their use as a phenology threshold in CLM5.0.

For our ABZ deciduous phenology algorithm, we therefore allow photosynthesis to begin when the following environmental criteria occur:

1. the 10-day average soil temperature in the 3rd soil layer is above 0° C

2. the 5-day average minimum daily 2m temperature average is above 0° C

3. when only a thin layer of snow remains on the ground ($< 10$ cm).

Together, these metrics approximate when plants begin photosynthesis in spring, allowing for roots to uptake moisture in unfrozen soil, for air temperatures to be consistently above freezing, and for plants to no longer be covered in snow.

### 2.4.2 Phenology Offset

Fall deciduous phenology in CLM5.0 is based on the same study focused on the temperate latitudes (White et al., 1997). As with phenology onset, biases arise from the extrapolation of temperate zone relationships to the high latitudes. Using NDVI,
senescence was identified to occur in autumn when daylight decreases ~11 hours. This daylight threshold is then set to be a global constant in CLM5.0. Complete dormancy is reached after 30 days after this photoperiod threshold. In the ABZ, this threshold of 11 hours of total daylight generally causes plants to decrease productivity in October and to begin dormancy in November. In reality, vegetation should be reaching dormancy at the end of September in the high Arctic (Zhang et al., 2004), with senescence beginning in August (Corradi et al., 2005). Based on existing physiological studies of ABZ vegetation,

it is unclear if temperature or photoperiod are the driving factor that triggers fall senescence (Marchand et al., 2004; Eitel et al., 2019), or if a combination of both is necessary for ABZ senescence (Oberbauer et al., 2013). Therefore, we focus on photoperiod, which is seasonally more consistent across the ABZ and clearly crucial for photosynthesis. Based on observations at high latitudes, 15 hours is a more accurate timing for senescence above 65° N (Corradi et al., 2005; Eitel et al., 2019). We scale the photoperiod threshold linearly along a latitudinal gradient from 65° N until ∼11 hours at 45° N such that the temperate

latitudes retain the offset timing determined by White et al. (1997).

### 2.4.3   Day Length Scaling for Photosynthetic Parameters

The Farquhar model of photosynthesis for C3 plants uses two main parameters to represent photosynthetic capacity, $J_{max}$ (the maximum rate of photosynthetic electron transport) and $V_{cmax}$ (the maximum rate of Rubisco carboxylase activtiy). In the current release of CLM5.0, $J_{max}$ and $V_{cmax}$ are predicted by a mechanistic model of Leaf Utilization of Nitrogen for

Assimilation (Ali et al., 2016, LUNA). Unlike previous versions of CLM, both $J_{max}$ and $V_{cmax}$ are prognostic in CLM 5.0, which allows for the vegetation to adjust to nutrients and environmental conditions. In our comparison of productivity in CLM5.0, we find that the prediction of $J_{max}$ and $V_{cmax}$ may be biased high in the ABZ (Rogers et al., 2017) when using algorithm values and schemes more appropriate for the tropics and temperate regions, which contributes to the overestimation of GPP by CLM5.0 across the ABZ.

Currently, $J_{max}$ is scaled in LUNA using day length:

$$f(daylength) = \left(\frac{daylength}{12}\right)^2 \tag{1}$$

The function, $f(daylegth)$, is a scaling factor that is based on the formulation in Bauerle et al. (2012), which quantifies the relationship between day length and $J_{max}$. However, the denominator in this equation in CLM5.0 is set to 12 hours, when it should be the maximum day length possible at a particular latitude (Bauerle et al., 2012). While 12 hours is fairly representative

for lower latitudes, this scale factor does not work for the ABZ where some regions experience up to 24 hour day light in summer, which allows f(daylength) > 1 in Equation 1, particularly around the summer solstice in June. To address this, we replace the default denominator of 12 hours with the geographically specific annual maximum hours of daylight that occur for a given grid cell.

### 2.4.4   Temperature Acclimation

Within both the Photosynthesis and LUNA schemes in CLM5.0, $J_{max}$ and $V_{cmax}$ are scaled from their values on the environmental leaf temperature and to 25° C using a modified Arrhenius temperature response function from (Kattge and Knorr, 2007, Supplement Fig. S3). Currently, $J_{max}$ and $V_{cmax}$ are allowed to acclimate to the plant's growth temperature, defined as the 10-day average 2-m temperature. However, the temperature acclimation function is limited to temperatures between between 11° C and 35° C and tuned to mostly temperate species (Kattge and Knorr, 2007). At temperatures outside of the acclimation

range, the temperature acclimation function scales $J_{max}$ and $V_{cmax}$ to unusually high values (Supplement Fig. S3), likely due to the use of temperate species for parameterization tuning. The mean daily summer temperature in the ABZ above 60° N is

below 11° C (Kalnay et al., 1996, NCEP/NCAR reanalysi)s), which implies vegetation at this latitude may never enter the range for temperature acclimation designated by Kattge and Knorr (2007). The temperature scaling done below 11° C is not based on any ABZ studies, nor does it match the previous scaling used in CLM5.0 parameterizations from Leuning (2002), which do contain some field sites in the ABZ. At more southern locations in the ABZ, vegetation may fluctuate around this minimum threshold value of 11° C, allowing discontinuities to appear in the temperature scaling of $J_{max}$ and $V_{cmax}$ and influencing biases in the seasonality of $CO_2$ fluxes. Due the lack of observational data across the ABZ incorporated in the Arrhenius function for acclimation in Kattge and Knorr (2007) we choose to implement temperature scaling functions from Leuning (2002), which does not create a discontinuity in $J_{max}$ and $V_{cmax}$ at such a critical temperature for the ABZ. This is a standard implementation of the Arrhenius temperature response function, which has been shown to work well at lower temperatures under present climate conditions in previous versions of CLM. The shift from Leuning (2002) to Kattge and Knorr (2007) was originally made due to improved process understanding of acclimation in photosynthesis, but Kattge and Knorr (2007) would only be suitable if ABZ sites had been included in the parameterization. Without those sites, it is an extrapolation of the southern scheme to the ABZ, which we find introduces significant biases.

### 2.4.5   $J_{max}$ and $V_{cmax}$ Winter Default

The LUNA module calculates $J_{max}$ and $V_{cmax}$ dynamically during the growing season only. When plants are dormant in winter (non-growing season), CLM5.0 uses constant values (See equations in the Supplement Section 4). Thus, at the start of the growing season (or first day of spring), $J_{max,t}$ and $V_{cmax,t}$ are directly calculated from a winter constant value:

$$J_{max,\text{last day of winter}} \quad = \quad J_{max,t-1} = 50 \tag{2}$$

$$V_{cmax,\text{last day of winter}} \quad = \quad V_{cmax,t-1} = 85 \tag{3}$$

We find that this global default winter value strongly influences the prediction of $J_{max}$ and $V_{cmax}$ throughout the entire growing season (Supplement Fig. S4). In all of the ABZ PFTs, raising these default values increases mean growing season GPP, whereas decreasing them lowers GPP (Supplement Fig. S4). Furthermore, the constant winter values in Equation **??** represent a high bias globally in $V_{cmax}$ (Lawrence et al., 2019), contributing additional bias. Due to the sensitivity of this choice and in an effort to leverage the physiological history of a given location, we choose to save the average predictions of $J_{max}$ and $V_{cmax}$ from the previous growing season for all PFTs ($J_{max,prevyr}$ and $V_{cmax,prevyr}$). We preserve the LUNA equations, except these constant values.

$$J_{max,pft,\text{last day of winter}} \quad = \quad J_{max,pft,prevyr} \tag{4}$$

$$V_{cmax,pft,\text{last day of winter}} \quad = \quad V_{cmax,pft,prevyr} \tag{5}$$

### 2.4.6   Carbon Allocation

Finally, we investigate the sensitivities of parameters related to carbon allocation, which are relatively uncertain and strongly influence $CO_2$ fluxes in the ABZ, particularly the stem-to-leaf ratio and the root-to-leaf ratio. In CLM5.0, the parameter defining

the root-to-leaf allocation ratio is set at a constant value of 1.5 for all PFTs. This is not an ideal configuration as boreal trees and tundra vegetation are structurally different than other plant types due to the need to cope with colder temperatures, which should be reflected in allocation to their roots, leaves, and other plant components. Even within a PFT, different species have been measured to have drastically different ratios of allocation to roots and leaves (Iversen et al., 2015), but for the purpose of circumpolar simulations, we limit our allocation parameters to the PFT level. Root to leaf ratios have been measured as consistently high for grasses and shrubs, meaning more allocation to roots than leaves (Chapin III, 1980; Iversen et al., 2015), as the large root systems are key to survival of these Arctic species (Archer and Tieszen, 1983). We, therefore, tested a higher root-to-leaf allocation of 2 for shrubs and grasses, which agrees relatively well with observations of tundra vegetation (Buchwal et al., 2013). For boreal trees, below ground allocation in evergreen conifers has been found to be higher than in deciduous trees (Gower et al., 2001; Kajimoto et al., 1999). Lowering the root-to-leaf ratio of DBT to 0.75 better represents the typically shallow root systems of deciduous boreal trees (Kobak et al., 1996), while being consistent with values implemented in other deciduous tree modeling studies (Arora and Boer, 2005). Observations suggest that boreal NETs in general have more extensive root systems than the deciduous trees (Gower et al., 1997), thereby requiring more belowground resources, and that tundra shrubs and grasses allocate even more photosynthate belowground. Thus, observations provide support for NET root-to-leaf allocation to be larger than DBT, and we choose to allow the NET root-to-leaf allocation to remain at the CLM5.0 default value.

Regarding stem allocation, CLM5.0 includes an option for dynamic stem-to-leaf allocation. The ratio is based on NPP and can be used for woody trees and shrubs (Friedlingstein et al., 1999), generally acting to increases woody growth in favorable conditions (Vanninen and Mäkelä, 2005). Though this allocation was previously used in CLM4.5 and turned off for CLM5.0, it is an uncertain choice as noted explicitly in Lawrence et al. (2019). A problem with this allocation scheme was noted in the tropics (Negrón-Juárez et al., 2015), but not in temperate or high latitude climates. We test both options, comparing the static stem-to-leaf ratios used in CLM5.0 to the dynamic allocation option.

# 3 Results

We investigate the simulation of $CO_2$ fluxes in the arctic-boreal zone by CLM5.0 using gridded and point simulations. We identify biases in the carbon cycle in Section 3.1, present our mechanistic and additive improvements to phenology and photosynthesis in Section 3.2, and make our model recommendation in Section 3.3. This ABZ analysis focuses primarily on GPP fluxes in CLM5.0, but as expected, our assessment extends to TER and NEE due to the interdependence of these carbon fluxes on productivity (Chen et al., 2015). We assess simulation biases based on the comparison of CLM5.0 output against FLUX-COM and EC towers, as detailed in Section 2.2. We address the biases that arise from using the default CLM5.0 parameters and schemes described in Section 2.4, which can be classified as: (1) phenology onset, (2) phenology offset, (3) daylight scaling, (4) Leuning temperature scaling, (5) initial spring value of $J_{max}$/$V_{cmax}$, (6) dynamic stem-to-leaf carbon allocation, (7) realist root-to-leaf carbon allocation.

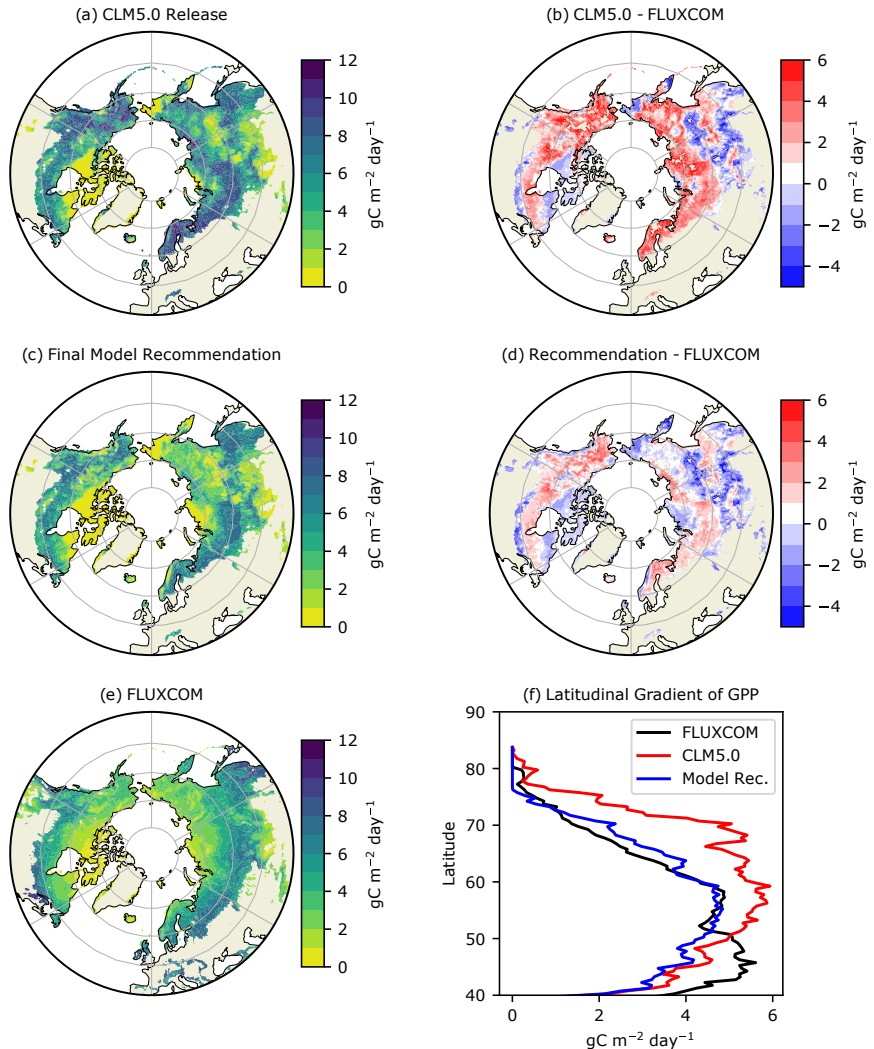

**Figure 1.** Average Summer (JJA) GPP (gC m$^{-2}$ day $^{-1}$) for (a) CLM5.0 Release, (b) our Model Recommendation, and (c) FLUXCOM, with the difference between model simulation and FLUXCOM in (b) and (d). The latitudinal gradient of summer GPP is depicted in (f).

 ## 3.1 Biases in CLM5.0

The latest release of CLM5.0 substantially overestimates summer GPP in the ABZ by $\sim$3 gC m$^{-2}$ day$^{-1}$ or 40% (red line in Figures 1b and 2a). The magnitude of this bias is such that CLM5.0 estimates of GPP for high latitudes tundra vegetation are comparable with the more southern boreal forests (Fig. 1a). This lack of a latitudinal gradient in CLM5.0 is not supported by FLUXCOM and ILAMB benchmarking (Fig. 1f). Though most of the ABZ in CLM5.0 is over productive, we note that there is a large area with $GPP = 0$ in Siberia, indicating that this region is not photosynthesizing in CLM5.0 though it should be. Thus, the simulation of carbon fluxes in CLM5.0 is very heterogeneous with areas that are highly productive and areas that are not-functioning, or "dead-zones".

We next investigate the seasonal cycle of $CO_2$ fluxes in the ABZ. Across the ABZ, average productivity in the CLM5.0 simulation is high throughout the year compared to FLUXCOM, which is shown throughout the year in Figure 2a. From a seasonal perspective, CLM5.0 vegetation enters dormancy later than observations, as can be seen by the high biases in GPP in fall. The timing of photosynthesis in spring appears accurate when we look at the PFT-aggregated average of $CO_2$ fluxes. We see a similar high bias in TER in Figure 2b, as respiration is tightly coupled to the high biased GPP. The magnitude of peak summer NEE in CLM5.0 matches observational data better than GPP and TER. However, its seasonal cycle exhibits biases and timing issues related to spring drawdown, summer minimum, and fall peak NEE (Fir. 2c).

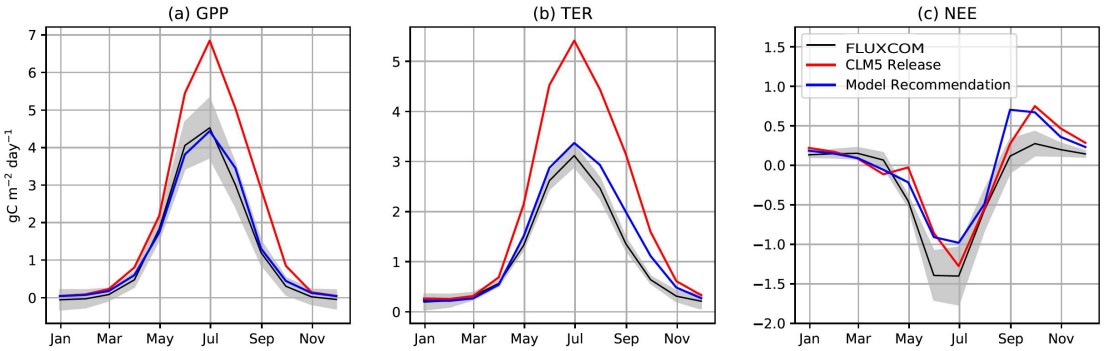

**Figure 2.** The annual cycle of (a) GPP, (b) TER, and (c) NEE from CLM5.0 and our Model Recommendation compared to FLUXCOM in the ABZ. Non-productive grid cells in the ABZ are removed from the average, meaning where LAI=0, which is standard procedure in CLM analysis (Lawrence et al., 2019).

Assessing the seasonality of $CO_2$ fluxes in the ABZ using the PFT-specific output of CLM5.0 reveals biases in phenology that are hidden when PFTs are aggregated together in a grid cell. In terms of phenology, we find that NET begins significant photosynthesis in February in CLM5.0 when air temperatures are well below freezing. This onset of NET photosynthesis is considerably early according to both FLUXCOM (Figure 3b) and our understanding of available liquid water for photosynthesis (Goulden et al., 1998). The peak productivity in NET occurs in June, instead of July as seen in observational data. In contrast, the onset of photosynthesis for deciduous trees and shrubs is consistently late (Fig. 3). The gridded CLM5.0 GPP output hides

these biases, showing that on average onset in CLM5.0 matches observations (Fig. 1a). In contrast, the high bias of CLM5.0 productivity during late autumn was easily seen in the gridded CLM5.0 output, and we confirm that this bias is due to the shifted seasonal cycle of deciduous PFTs (Fig. 3).

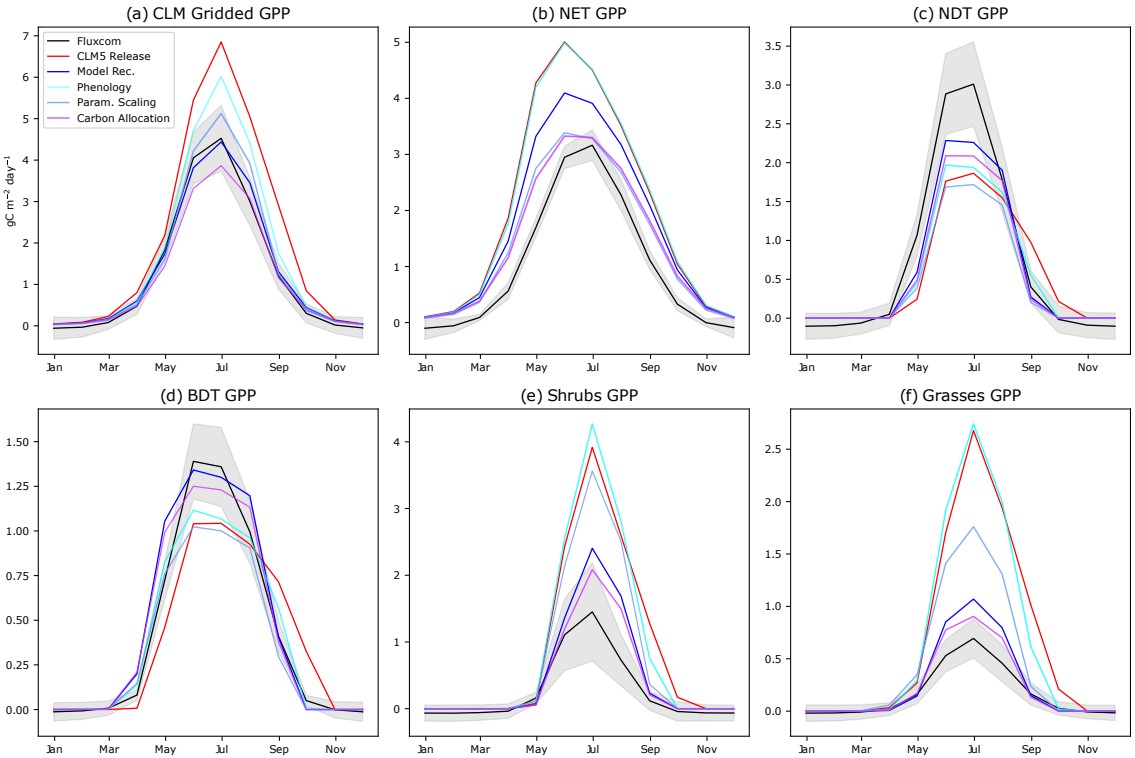

**Figure 3.** The annual cycle of GPP for CLM5.0 release with intermediate model development steps compared to FLUXCOM for: (a) aggregated gridded CLM5.0 output, (b) NET, (c) NDT, (d) BDT, (e) shrubs, (f) grasses. Phenology incorporates changes to onset/offset. Param. Scaling adds our scaling of $J_{max}$ and $V_{cmax}$ by daylight and Leuning. Carbon allocation adds changes to the root-to-leaf and stem-to-leaf parameters. The final model recommendation incorporates the bug fix (Supplement Section 3) and spring initialization of $J_{max}$ and $V_{cmax}$.

Regarding the magnitude of photosynthesis, the CLM5.0 PFT-specific output indicates that both shrubs and grasses have a large high bias of GPP compared to observational data by a factor of 2-3 (Fig. 3a). We confirm that the tundra grass and shrub PFT specific output is often greater than or as productive as the boreal trees in CLM5.0 (Fig. 3 e,f vs. b,c,d). The deciduous boreal trees (NDT and BDT) have a low growing season GPP bias, while NET have a high productivity bias. By examining the spatial pattern of the average summer GPP (Fig. 1), one can see that there are many areas where $GPP = 0$ for many consecutive years, indicating that the PFT did not survive. In Siberia, a prominent "dead zone" occurs in what should be highly productive deciduous needleleaf larch forests. Smaller "dead zone" areas are present for all other PFTs across the ABZ (Supplement Fig. S5).

As with the aggregated CLM5.0 output, the PFT specific biases in TER are similar to those noted for GPP (Supplement Fig. S6). PFT-specific patterns in NEE also tend to follow the biases in GPP and TER, with the notable exception of spring in deciduous vegetation. For these PFTs, there is a large spike of $CO_2$ release to the atmosphere, up to 0.5 gC m$^{-2}$ day$^{-1}$ between

April and May that does not match observations (Supplement Fig. S7). This is primarily due to the late onset photosynthesis at a time when TER is increasing due to warming air and soil temperatures. We note that the net balance of BDTs is near 0, rather than a sink, which agrees with our conclusions that deciduous trees are not productive enough relative to FLUXCOM and flux towers. Ultimately, the timing and magnitude of biases in TER and NEE confirm our need to focus on GPP at a first step in better representing seasonal CO2 fluxes in CLM5.0 for the ABZ.

Benchmarking CLM5.0, yields the following primary issues for the simulation of GPP across the ABZ:

1. The onset of GPP in deciduous PFTs in spring is consistently late across all PFTs.

2. The fall senescence of GPP is consistently late.

3. There is no latitudinal gradient in summer GPP, due in part to the high GPP bias in tundra grasses and shrubs.

4. NET begin photosynthesis early in winter and reach peak in productivity in June, instead of July.

5. NET have a high GPP bias throughout the growing season.

6. Deciduous trees (BDT and NDT) have a low GPP bias.

7. There are large areas of PFTs that have no productivity at all or are effectively dead in CLM5.0, particularly the NDT in Siberiak, representing larch forests (larix spp.).

## 3.2 Model Development at Flux Towers

We confirm these biases in the ABZ by comparing the CLM5.0 PFT-specific output to representative flux towers across the ABZ. For example, the southern boreal mixed forest site at CA-QC2 (Figure 4a) contains NET, BDT, and shrub PFTs. The BDT here is a dead zone, where $GPP = 0$, making it a useful site to understand what model thresholds may be influencing one PFT to die out in a simulation, while others remain productive. We also include a BDT site at CA-OAS (Figure 4b) to further investigate the simulation of deciduous trees at an alive grid cell in CLM5.0. The grasses and shrubs at Eight Mile Lake, Alaska are approximately five times more productive than observations (Figure 4c). In contrast, at the larch forest at RU-SKP (Figure 4d), we find that onset is late and GPP is roughly half the observed value, consistent with the low bias in NDT gridded output. We perform model development by examining each issue described in Section 2.4 sequentially on our model development sites: CA-QC2, CA-OAS, US-EML, and RU-SKP. Each flux tower measurement is carefully chosen to cover all CLM5.0 PFTs to understand their complex impacts on GPP across all PFTs, while also including a "dead-zone" to tease out a different kind of bias. Model development follows the procedure and biases described in Section 2; we begin with phenology, move on to the photosynthesis schemes, and end with adjusting the carbon allocation parameters. The use of flux towers and point simulations allow us to test a range of hypotheses for each model development objective. Successful model development

is achieved when GPP is within standard deviations limits. We choose to make the model development additive because the improvements are generally small and justified observationally, but all together make for a substantially improved simulation of ABZ carbon fluxes (Fig. 3).

Regarding phenology, using new thermal and moisture thresholds for onset, the deciduous plants to begin photosynthesis earlier in the season, which more closely matches observations (Fig.3). However, the bias in the magnitude of photosynthesis is not improved by our phenology changes; In fact, the productivity of grasses and shrubs increases further with a growing season that begins earlier in Spring (Fig. 3e,f, comparing the red "CLM5.0 Release" line to the cyan "Phenology" line).

Next, regarding $J_{max}$, we find that modifying the function that scales $J_{max}$ to accurately use the maximum number of daylight hours on each grid cell (Bauerle et al., 2012) decreases productivity across all PFTs (Figure 4 and Supplement Fig. S8). In particular, we decrease the June spike in GPP for NET because the daylight fraction around the summer solstice is no longer greater than 1. Overall, this modification decreases the high bias in ABZ GPP by 2 gC/m$^2$ in the summer (Supplement Fig. S8) and generates a latitudinal gradient in the PFT specific output of trees (Supplement Fig. S5). By reverting the CLM5.0 temperature scaling scheme from (Kattge and Knorr, 2007) to Leuning (2002) as used on CLM4.5, we find that GPP is decreased for all PFTs, especially in spring when the photosynthesis ramp up had been artificially high. Our model updates improve phenology of NETs due to more realistic scaling of $J_{max}$ and $V_{cmax}$ (Fig. 3b, the light blue "Temp Scaling" simulation and Fig. S12). The GPP of grasses is also decreased (Fig. S13), but shrubs are still biased high.

After decreasing productivity of all PFTs (Fig. 3, compare the red "CLM5.0 Release" with the blue "Param. Scaling"), tundra shrub and grass productivity in CLM5.0 retain a substantial high GPP bias (Fig. 3e,f), while the deciduous tree PFTs have a low GPP bias (Fig. 3c,d), which may be related to non-optimized ABZ carbon allocation parameters. Allowing for a dynamic stem-to-leaf allocation improves both the timing and magnitude of photosynthesis (Supplement Fig. S9). Additionally, we make PFT specific alterations to root-to-leaf ratios based on our findings from Section 2.4. As a result, GPP is lowered in grasses and shrubs and increased for deciduous trees (NDT and BDT), approaching FLUXCOM values (Fig. 3). The rest of our changes to the model involve schemes in LUNA that we believe are initialized incorrectly for the ABZ, but also the rest of the world. These recommended model updates include initializing the winter default values of $J_{max}$ and $V_{cmax}$ using the mean value for a given grid cell during the previous growing season (Equations 4,5 and Supplement Fig. S4) and a model error related to the calculation of 10-day leaf temperature (Supplemental Section S3).

### 3.3 Improved Carbon Fluxes in the Model Recommendation

Based on our changes to phenology, photosynthesis, and carbon allocation schemes, we recommend the following mechanistically based changes to CLM5.0 for a considerably improved representation of CO2 fluxes in the ABZ:

1. GPP onset is based on soil temperature, air temperature, and snow cover.

2. GPP offset is based on a latitudinal photoperiod gradient such that the high Arctic begins senescence earlier.

3. $J_{max}$ is scaled by maximum day length instead of a constant 12 hours.

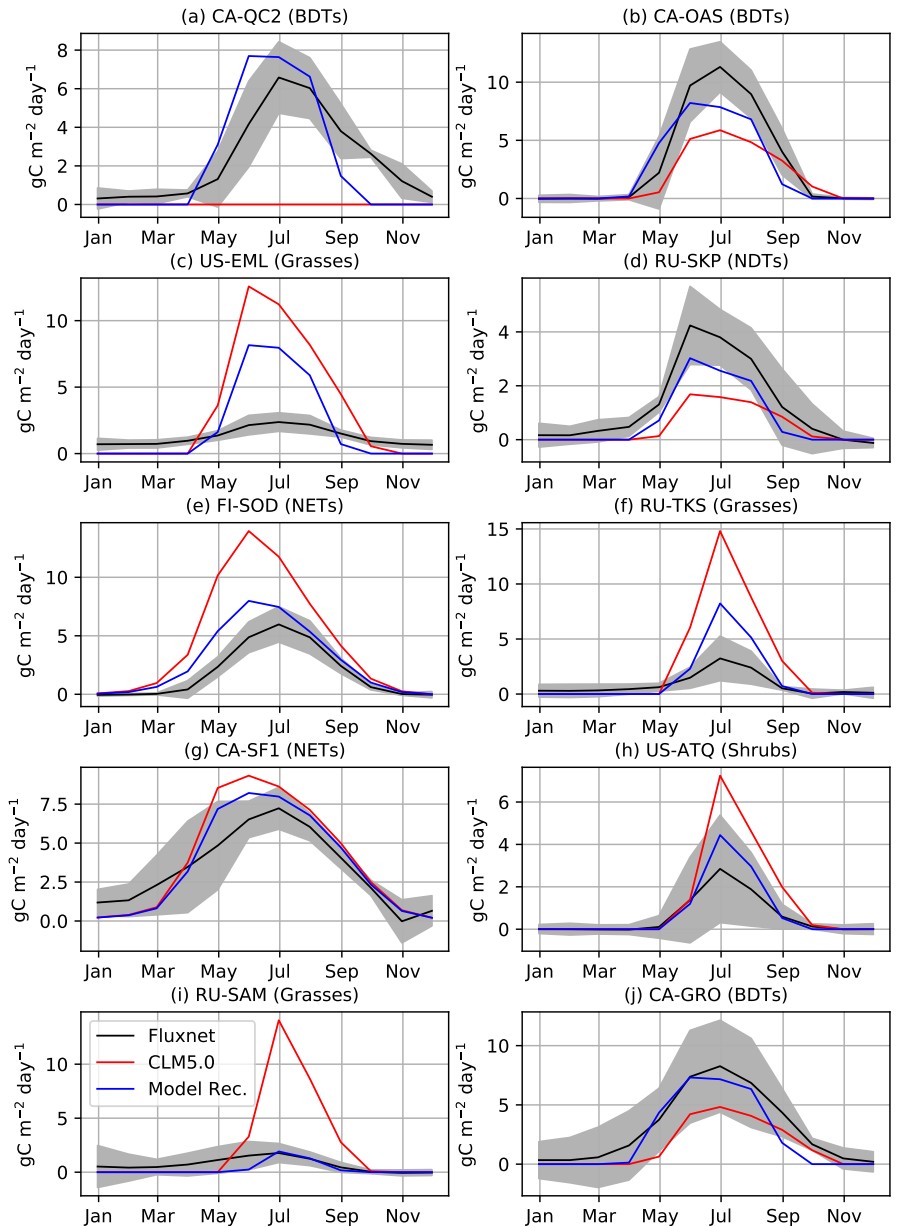

**Figure 4.** Comparison of GPP at Flux Tower sites to the CLM5.0 release and the results of our model development. We performed model development at CA-QC2 (mixed forest site in Quebec), CA-OAS (Aspen forest in Saskatchewan), US-EML (Eight Mile Lake Tundra location), and RU-SKP (Yakutsk Spasskaya Larch Forest). We did additional comparisons with FI-SOD (Sodankyla, Finland), RU-TKS (Tiksi grasslands), CA-SF1 (Saskatchewan boreal forest), RU-COK (Chokurdakh Tundra shrubs), RU-SAM (Samoylov grasslands), and CA-GRO (Groundhog River Boreal Forest).

4. $J_{max}$ and $V_{cmax}$ are scaled by temperature response functions parameterized by Leuning (2002).

5. $J_{max}$ and $V_{cmax}$ have initial values in spring that are based on the LUNA predictions from the previous growing season, allowing the values to vary across PFTs, time, and space.

6. The stem-to-leaf carbon allocation ratio for trees and shrubs is allowed to be dynamic throughout the season.

7. Observationally based root-to-leaf carbon allocation ratios are used. For deciduous tree PFTs, the root-to-leaf ratio is decreased, while the ratio for shrubs and grasses is increased.

We first validate that our modifications to CLM5.0 offer improvements to simulations of $CO_2$ fluxes when looking at specific flux towers (Fig. 4). For instance, productivity is increased at BDT at CA-OAS, which is further validated at CA-GRO. The "dead zone" at CA-QC2 is now highly productive and much closer to observed carbon fluxes in terms of seasonality and magnitude. The new cold deciduous onset scheme causes this improvement, as the previous growing degree day scheme in CLM5.0 prevented photosynthesis from ever starting. Photosynthesis in NDT is increased due to our model development, but as shown in both gridded output and RU-SKP, the NDT photosynthesis needs to increase further. The phenology and magnitude of NET is much improved at validation sites. However, as is expected (Schaefer et al., 2012), not all of the flux tower sites have $CO_2$ fluxes that are reproduced within the range of observational uncertainty (Fig. 4c and Supplement Fig. S11). For example, although GPP at US-EML is reduced due to our model development, it is still biased high. However, the grasses and shrubs at other validation sites are much improved compared to flux measurements, like the grasses and shrubs at RU-SAM, RU-TKS, and US-ATQ.

We next compare our model improvements in a gridded simulation against $CO_2$ fluxes from FLUXCOM (Figure 2a). The high productivity bias at high latitudes is substantially reduced due to our model development by decreasing the productivity of ABZ shrubs and grasses. Our Model Recommendation for CLM5.0 produces a latitudinal gradient (Fig. 1f) in productivity, with the tundra no longer being as productive as the boreal forest (Fig. 3). The timing of photosynthesis is also improved as dormancy is reached by October in most PFTs, instead of November (Figs. 1 and 3). In examining the PFT specific output (Fig. 3), we confirm an earlier beginning to spring photosynthesis in deciduous trees, shrubs, and grasses. In contrast, our model development successfully delays the onset of productivity in NET, due to a combination of daylight scaling and the temperature scaling from Leuning (2002). As for the magnitude of photosynthesis, we find that NET photosynthesis is reduced across the ABZ, while the deciduous boreal tree PFTs experience an increase in productivity (Supplement Fig. S10).

Although GPP is our main focus of development due to the number of productivity biases, we find improvements in other component $CO_2$ fluxes. Changes in TER generally follow those of GPP, and our modifications substantially reduce the high summer respiration bias in CLM5.0 (Fig. 2), which is mostly due to the reduction of GPP and TER in grasses and shrubs at high latitudes(Supplement Fig. S6). The respiration of deciduous tree PFTs did not change substantially, but TER decreases in NET due to the decrease in GPP. Due to the improvements in GPP onset/offset, the model no longer simulates large net $CO_2$ emissions in spring (Supplement Fig. S7), which better matches observations of NEE. In many PFTs the respiration and thus NEE in fall are high in the gridded output. Although this does not match FLUXCOM, it is consistent with recent measurements

of high fall respiration in the ABZ (Commane et al., 2017; Natali et al., 2019). The net balance of carbon fluxes does not change substantially due to our model development. We find that the carbon sink in ABZ CLM5.0 decreases by about 12%, which is due to a smaller net sink in the summer and larger carbon release in fall.

## 3.4 Validation with ILAMB

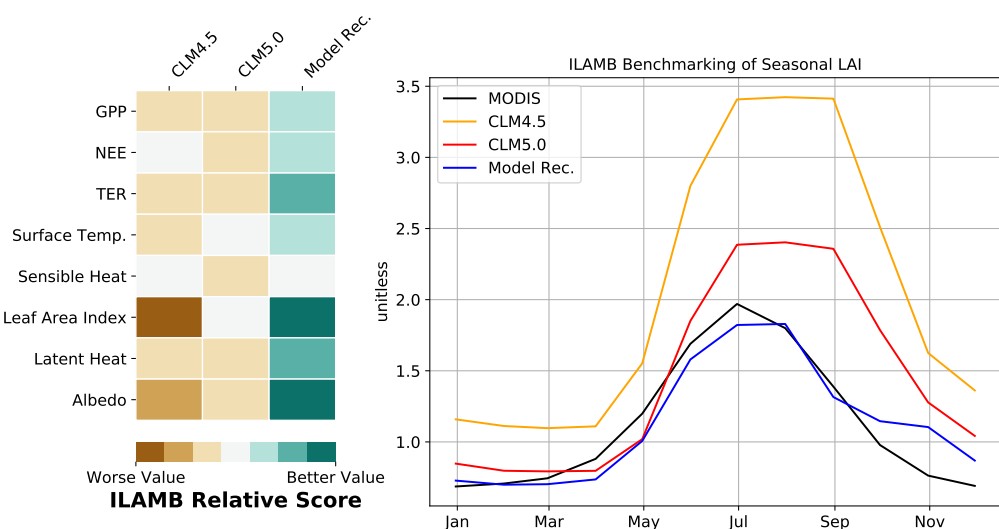

**Figure 5.** ILAMB benchmark score of CLM5.0 Model Development Recommendation against previous versions of CLM: 5.0 and 4.5 (left). LAI seasonal results compared to MODIS (right).

Comparing output from CLM5.0 and our Model Recommendation to observational data provided by the ILAMB framework confirms broad improvements in model fidelity (Fig. 5). This includes the $CO_2$ fluxes we focus development on, as well as surface fluxes (sensible heat, latent heat, albedo). ILAMB confirms that the high GPP bias and late phenology biases are reduced. LAI, in particular, has been improved greatly in the Arctic in regards to both timing and magnitude (Fig. 5). According to the ILAMB score, the implemented changes are not detrimental to any other essential land surface variables, and in fact improve their simulation according to the centralized benchmark scores. Breaking down the overall benchmark score, our Model Recommendation has large improvements in the spatial distribution and interannual variability scores. The seasonal cycle score of GPP did not improve substantially, which make sense due to the phenology problems only becoming apparent in the PFT specific output, which is not a standard ILAMB benchmark. The NEE seasonal cycle is substantially improved according to ILAMB, which agrees with our FLUXCOM validation. The relative improvements to moisture and heat fluxes are particularly noteworthy, as these changes can feed back to the regional climate system.

We are also interested in contributing to the improvement of the global CLM5.0 simulation. We confirm that a global simulation is possible and reasonable (Fig. 6) with these additions to the code base. All but two of our model improvements are

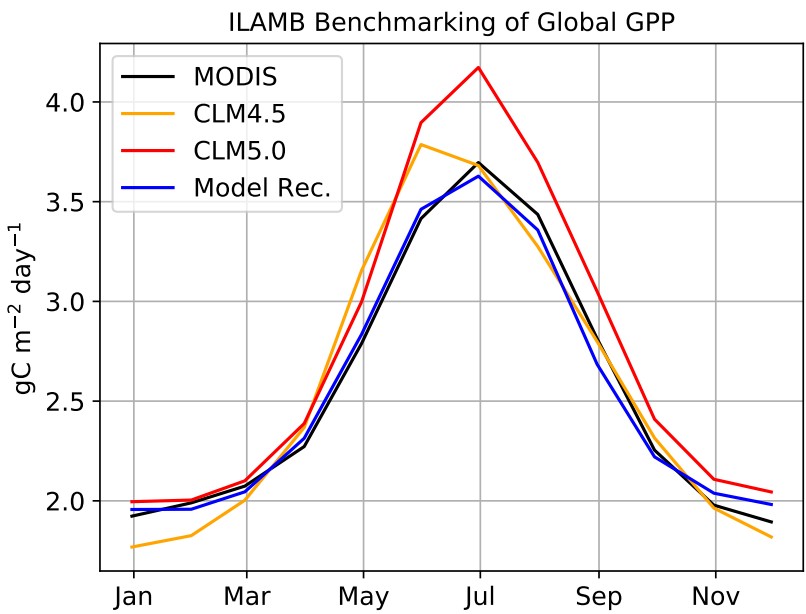

**Figure 6.** Seasonal Global GPP Benchmakred in ILAMB. Our model recommendation for the ABZ is applied globally and benchmarked for GPP in CLM4.5 and CLM5.0.

limited to the ABZ, meaning that we do not expect significant biases to emerge at lower latitudes due to our model development, and our ILAMB validation confirms this. The constant LUNA equation is one of the global changes and the other is the change in temperature scaling from Kattge and Knorr (2007) to Leuning (2002). Additional testing at lower latitudes, which is outside of the scope of this study, is necessary to determine the effects on the global carbon budget.

## 4   Discussion

Through mechanistic model development, we have reduced the biases in carbon cycling in CLM5.0 for the Arctic-Boreal PFTs. Many of our recommendations in Section 2.4 affect several of the biases noted in Section 3.1, indicative of the many interconnected schemes in CLM5.0. Ultimately, we believe our modifications to be reasonable, based in observations, and

a step towards a more accurate simulation of carbon cycling in high latitude terrestrial ecosystems, but we also discuss the limitations of our model development choices and identify avenues of research that could continue to improve CLM5.0.

Having more accurate phenology in CLM5.0 is critical for understanding the recent and future changes in biogeochemistry in the ABZ, as global change drivers during the shoulder seasons may be drivers of carbon cycle changes. Deciduous trees have protective mechanism to avoid the onset of growth when there is a strong probability of cold snaps (McMILLAN et al., 2008), and our new thresholds are key environmental conditions designed to mimic the end of winter signals. The use of threshold values in onset schemes is a well established phenology method (Jolly et al., 2005; Arora and Boer, 2005) that is used in other

models, like LPJ (Forkel et al., 2014) and the Canadian Terrestrial Ecosystem Model. These schemes can be fairly simple or more complicated, much like growing degree day models (GDD). We choose to focus on environmental thresholds due to the uniqueness of the ABZ environment, which is dominating by freeze-thaw dynamics, making a threshold approach effective. We did investigate other GDD schemes, but we found that many GDD models perform similarly (Hufkens et al., 2018) and in the ABZ have a documented late onset (Botta et al., 2000; Fu et al., 2014), just like the current scheme in CLM5.0. We did not

identify a scheme well validated in the ABZ, as most observational validation studies are focused on the lower latitudes, and those studies also identify that gridded phenology products tend to be produced using optical imagery, which often does not correspond well with CO2 fluxes in shoulder seasons (Fisher et al., 2007; Parazoo et al., 2018a). Ultimately, using another GDD scheme from the known literature would be trading one extrapolated temperate scheme for another. This threshold approach may have limitations for a warming world, which is why an ABZ focused phenology study, using novel datasets like PhenoCam

(Richardson et al., 2018; Hufkens et al., 2018), have potential to uncover more complicated vegetations processes and filling in gaps from satellite based studies (Fisher et al., 2007).

    Implementing an offset scheme with a latitudinal gradient in CLM5.0 is a first step towards more realistic timing of fall senescence in CLM5.0, and additional work is needed to understand how temperature and other climate drivers impact the timing of dormancy. Studies are divided on the issue of whether temperature or photoperiod are driving offset in the ABZ.

Field experiments have found photoperiod to be a likely driver at high latitudes (Eitel et al., 2019), but temperature and even precipitation may be more important at lower latitudes. Experiments of high arctic tundra have shown that senescence can be delayed by up to 15 days due to warming (Marchand et al., 2004). Given this uncertainty, we suggest that photoperiod is a sufficient and justified approximation for fall dormancy, until better mechanistic relationships can be derived from observational data. The late bias in fall phenology has also been noted in other TBMs (Richardson et al., 2012), also likely due to extrapolation

of temperate schemes to the high latitudes. Thus, our simple daylight threshold here could be applicable to other models.

    One of the most impactful changes to CLM5.0 GPP is the use of maximum day length to scale $J_{max}$, rather than 12 hours. Equation 1 would previously scale the prediction of $J_{max}$ high, particularly in June with the summer solstice and day length at its maximum. The previous use of 12 hours for maximum day length is likely a holdover from using LUNA in the tropics, as (Bauerle et al., 2012) was the cited basis for Equation 1, which used maximum day length appropriately. After we fix this

substantial scaling bias, we find other algorithms and parameterizations are more sensitive to model changes. This opens up many avenues of model development, some of which we are able to accomplish in this study, like more realistic allocation

parameters and scaling changes to $J_{max}$ and $V_{cmax}$. For future steps, we argue that a re-parameterization CLM is necessary, as previous model tunings attempted to bring down the high GPP bias through parameter choices rather than this bug fix.

As with the length of day scaling described above, the photosynthetic temperature acclimation in scaling of $J_{max}$ and $V_{cmax}$ were not created for ABZ latitudes and tend to exacerbate model biases. We, therefore, recommend not using these functions from (Kattge and Knorr, 2007), and stress the need for further research on photosynthetic temperature acclimation in the ABZ, especially for projecting responses to future climate. The current implementation generates unrealistic seasonal temperature response functions for GPP resulting in model biases (Smith et al., 2017). Previous work by Rogers et al. (2017) advocated for removing the 11° C limit from (Kattge and Knorr, 2007), but our tests of this did not decrease productivity of the grasses and shrubs at high latitudes (not shown) or offer any other improvements, at least not without a re-parameterization of the acclimation scheme. We do advise caution in using Recently, Kumarathunge et al. (2019) have created an acclimation parameterization that included ground measurement sites from Utqiagvik (Barrow), Alaska and Finland. Though most of Canada and Siberia are not represented in the parameterization, including this recent observationally-based acclimation function in CLM5.0 is a logical next step for a better ABZ simulation that includes temperature acclimation, a critical process for projecting carbon budges into the future.

The initialization of $J_{max}$ and $V_{cmax}$ in spring is a highly sensitive choice (Supplement Fig. S4). We find improvements in GPP when using the mean predicted value of $J_{max}$ and $V_{cmax}$ from the previous year as an initial spring value. The default values for spring $J_{max}$ and $V_{cmax}$ in CLM5.0 are high for ABZ PFTs (Lawrence et al., 2019). Now with a default value that considers the PFT and climatological conditions of $J_{max}$ and $V_{cmax}$, the simulated seasonal cycle of $J_{max}$ and $V_{cmax}$ mimic the timing of GPP more closely. We observe only very small temporal fluctuation in these average values of $J_{max}$ and $V_{cmax}$, indicating that the LUNA predictions are relatively stable for each geographic-climatological region. The productivity for most ABZ PFTs is decreased throughout the whole summer due to this change in CLM5.0. Though we note that LUNA simulates lower values of $J_{max}$ and $V_{cmax}$ in most Arctic-boreal PFTs, this implementation of average $J_{max}$ and $V_{cmax}$ allows for spatial and temporal variability, including increases of $J_{max}$ and $V_{cmax}$ in some locations. Therefore, there is potential for this scheme to improve future predictions, as the sensitive initialized spring values can now adjust in a warming climate.

Finally, we examine the carbon allocation parameters in CLM5.0, which are a considerable and long-standing source of uncertainty in TBMs. Franklin et al. (2012) appropriately called carbon allocation the "Achilles' heel of most forest models". We argue that CLM5.0 does not use ideal carbon allocation values for the ABZ, but there are multiple diverging development paths for carbon allocation (Fisher et al., 2019) that could lead to a better simulation. During model development and testing of carbon allocation ratios for roots, leaves, and stems, we attempt to balance static and dynamic allocation schemes, but acknowledge there is always room for further development and improvement. The default parameters in CLM5.0 do not utilize dynamic stem-to-leaf carbon allocation due to exponential increases in biomass in the tropics (Negrón-Juárez et al., 2015), but not necessarily for the ABZ. We find that simulations of productivity are improved across all ABZ PFTs when this ratio is dynamic. In the model, the previous year's NPP is used to set this ratio, based on the assumption that NPP can serve as a proxy for environmental conditions that send resources to either roots or leaves, thereby increasing woody allocation in favorable growth environments (Vanninen and Mäkelä, 2005). In the previous version of CLM, CLM4.5, dynamic stem-to-

leaf allocation generally lowered carbon allocation, sending more carbon to leaves than stems, agreeing with observational comparisons (Montané et al., 2017), which is the same pattern we see in ABZ PFTs in CLM5.0.

The final improvements to the deciduous tree PFTs, grasses, and shrubs come from changing the root-to-leaf ratio, which is currently the same constant value for all PFTs and does not make sense ecologically (Iversen et al., 2015). We test a range of allocation values for each PFT, using observational data to guide our decisions. These static carbon allocation ratios are our best estimate for mimicking emergent plant processes, and we find that the allocation parameters relative to other PFTs may be more important for model simulations than matching observations exactly. The success of focusing on allocation parameters relative to other PFTs may be of guidance to other model schemes. The NDT allocation that we find does align with observations, is used in other models (Arora and Boer, 2005). The grass allocation parameter is low compared to some observations, but we find increasing the root-to-leaf ratio for grass PFTs in CLM5.0 only succeeds in killing productivity completely. We did consider changing the allocation of NETs, but had little success in finding parameters that made sense (Fig. S14). With the difficulty in relating allocation parameters to model counterparts, a more dynamic scheme may be needed for future model versions. For instance, the carbon allocation to roots may saturate when LAI=1 in Arctic shrubs (Sloan et al., 2013). Allowing TBMs to adjust these carbon ratios based on LAI, may be the next step in CLM5.0 model development, approximating the realistic behavior of plant species.

The "dead zones" in CLM5.0 are caused by a combination of issues, such as late onset and allocation parameters. However, these increases in productivity do not clearly and substantially improve the dead PFTs in CLM5.0 as a whole, particularly the non-productive area of Siberia (Fig. 1c). These areas of non-productivity are particularly problematic for future CLM5.0 simulations due to the suspected importance of Siberian $CO_2$ fluxes for current and future seasonal carbon balance (Zimov et al., 1996, 1999; Lin et al., 2020). Although we did not cause any additional areas to die, we also did not succeed in increasing productivity in the larch forests of Siberia. It is also worth noting that areas with "dead zones" clearly visible in the gridded product have all PFTs dead, not just NDT. We hypothesize that there may be thresholds for climatic drivers that inhibit photosynthesis. For instance, there is a minimum relative humidity threshold for nitrogen allocation in LUNA. This threshold appears to be somewhat arbitrary and the ABZ often experiences a dry continental climate. The larch forests of Siberia could benefit from a lower relative humidity threshold to raise their low productivity and potentially improve "dead zones". A test of this hypothesis does raise productivity for deciduous tree PFTs, but the "dead zones" do not become productive. We require more information on relative humidity and nitrogen allocation at low temperatures to determine if this parameter should be changed in the model. Here we focus on mechanistic model development, but a parameter specific to ABZ vegetation may lead to further improvements.

The relative improvements to each PFT are also significant. For instance, as noted previously, our model recommendation generates a latitudinal gradient in CLM5.0 (Fig. 1f), which is due to tundra vegetation being less productive. The Arctic grasses and shrubs are now less productive than the boreal trees, which is consistent with observations. The deciduous tree PFTs in our model recommendation are more productive than the CLM5.0 Release, with NDT productivity increasing by 20% and BDT increasing by 50% (not shown). Additional work is needed for the deciduous schemes, as deciduous trees are observed to be more productive than evergreen trees (McMILLAN et al., 2008). We did decrease the high bias of productivity in NETs by

20%, but NET are still causing a high bias in the simulation of GPP. The evergreen scheme has long been noted to be relatively simple in CLM5.0 (Lawrence et al., 2019) and a key next step for model development.

As with most studies of the high latitudes, we are limited by the availability of ground observations (Arora and Boer, 2005; Richardson et al., 2012). For instance, we only have one flux tower corresponding to NDT (RU-SKP), which is also unfortunately a PFT in CLM5.0 that needs substantial work, particularly in the "dead-zone" in Siberia (Supplement Fig. S5). For other PFTs we restrict our tower choices to have the best data available. We only include EC flux observations where the vegetation classes corresponded with the CLM5.0 PFTs that occur within the ABZ. Metadata indicating a mixed forest or a combination of short stature vegetation and trees causes comparisons to not be clean (Supplement Fig. S11), meaning we withhold these kind of sites for extra validation. These flux towers are not ideal sites for model development, but abz do generally confirm a reduction in GPP due to our model development efforts. The EC record must also span over at least three consecutive years before 2014 due to the forcing dataset used by CLM ending in 2014, which does not allow us to leverage the most recent ABZ ground observations. These criteria restrict our point-based analysis to only a few sites (Supplement Table S1), which does further limit our ability to make statistically robust conclusions. We attempt to mitigate this lack of data by validating carbon flux results with point simulations and gridded simulations. Then, we use ILAMB to confirm improvements in other independent variables that were no the focus of model development. More comprehensive model improvements for the ABZ may be possible through an increase in the availability and spatial representativeness of tower EC data. We look forward to the increased emphasis on data archiving, standardization, and synthesis, as well as more detailed examination of functional relationships and PFT-specific parameters. As our understanding evolves, observational networks improve, and long-term data archives grow, we stress the need for continued development and model fidelity for high latitude terrestrial ecosystems given their importance in the global climate system.

## 5 Conclusions

We have approached model development for $CO_2$ cycling in the ABZ in a mechanistic and targeted fashion, leveraging available observational data and derived products. We find that many biases are interconnected, meaning that mechanistic model development and bug fixes can improve part of the simulation, while making other aspects worse (Fig.3). Overall, our work with CLM5.0 in the ABZ highlights the importance of regional model analysis and development. We find the extrapolation of model schemes developed for temperate latitudes to the high latitudes to be the root of many biases, as has also been noted elsewhere (Rogers et al., 2017).

We find that a physically-based phenology formulation using soil temperatures, air temperature, and snow depth is more accurate than the existing parameterization developed for temperate latitudes. Allowing offset timing to vary with latitude instead of a single global value improved circumpolar leaf offset. We improve the scaling of day light in the maximum rate of electron transport ($J_{max}$) using the maximum day length. Additionally, we remove a global initialization of the maximum rate of electron transport ($J_{max}$) and the maximum rate of carboxylation ($V_{cmax}$) that biases prediction of these critical photosynthesis components each spring. We also recommend the Leuning (2002) temperature scaling of $J_{max}$ and $V_{cmax}$ for the ABZ, as

Kattge and Knorr (2007) is not optimized for the ABZ, biasing both the maximum productivity and phenology. Finally, we adjust carbon allocation ratios for ABZ PFTs to levels that better match observations and result in more realistic simulations of GPP.

Finally, we assess the performance of our model recommendation in a global simulation, confirming that a global simulation is possible and yields reasonable carbon fluxes. Furthermore, we actually identify improvements in the global carbon cycle and
610 budget according to ILAMB metrics. The reduction of biases in the ABZ carbon cycle has implications for future projections with models that overestimate GPP. The modifications we implement here illustrate that previous extrapolations of temperate or even tropical observations cause significant biases. We advocate for more regional ABZ focused development to ensure accuracy in the ABZ when implemented in global simulations, as the high latitudes are a critical component of the rapidly changing climate system.

*Code availability.*  Our model recommendations are currently on Github for incorporation into the CESM master branch and CLM5.1. Please see the pull request and discussion here: https://github.com/ESCOMP/CTSM/pull/947

A more permanent storage of the code is available here: http://doi.org/10.5281/zenodo.4706221

*Data availability.*  For any questions or interest in our model data, please reach out to the corresponding authors Leah Birch and Brendan Rogers.

*Author contributions.*  LB and BMR designed the study and model development. LB performed benchmarking, model development, and validation. All authors contributed to the manuscript.

*Competing interests.*  The authors declare no competing interests.

*Acknowledgements.*  This work was funded by the National Aeronautics and Space Administration (NASA) Arctic-Boreal Vulnerability Experiment (ABoVE) and Carbon Cycle Science programs (NNX17AE13G).
Funding for AmeriFlux data resources was provided by the U.S. Department of Energy's Office of Science. This work used eddy co-variance data acquired and shared by the FLUXNET community, including these networks: AmeriFlux, AfriFlux, AsiaFlux, CarboAfrica, CarboEuropeIP, CarboItaly, CarboMont, ChinaFlux, Fluxnet-Canada, GreenGrass, ICOS, KoFlux, LBA, NECC, OzFlux-TERN, TCOS-Siberia, and USCCC. The ERA-Interim reanalysis data are provided by ECMWF and processed by LSCE. The FLUXNET eddy covariance data processing and harmonization was carried out by the European Fluxes Database Cluster, AmeriFlux Management Project, and Fluxdata
project of FLUXNET, with the support of CDIAC and ICOS Ecosystem Thematic Center, and the OzFlux, ChinaFlux and AsiaFlux offices.

We would like to thank all involved in the ILAMB project, particularly Nathan Collier, for providing a valuable resource for the community.

We would like to acknowledge high-performance computing support from Cheyenne (doi:10.5065/D6RX99HX) provided by NCAR's Computational and Information Systems Laboratory, sponsored by the National Science Foundation.

Finally, we would like to thank our three anonymous reviewers and the editors at GMD for their helpful comments.

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
