# Peer review of "Addressing Biases in Arctic-Boreal Carbon Cycling in the Community Land Model Version 5"

_Geoscientific Model Development, 2020_

## Short Comment (SC1) · 21 Dec 2020

Dear authors,

in my role as Executive editor of GMD, I would like to bring to your attention our Editorial version 1.2:

https://www.geosci-model-dev.net/12/2215/2019/

This highlights some requirements of papers published in GMD, which is also available on the GMD website in the 'Manuscript Types' section: http://www.geoscientific-model-development.net/submission/manuscript_types.html

In particular, please note that for your paper, the following requirement has not been

met in the Discussions paper:

- Code must be published on a persistent public archive with a unique identifier for the exact model version described in the paper or uploaded to the supplement, unless this is impossible for reasons beyond the control of authors. All papers must include a section, at the end of the paper, entitled "Code availability". Here, either instructions for obtaining the code, or the reasons why the code is not available should be clearly stated. It is preferred for the code to be uploaded as a supplement or to be made available at a data repository with an associated DOI (digital object identifier) for the exact model version described in the paper. Alternatively, for established models, there may be an existing means of accessing the code through a particular system. In this case, there must exist a means of permanently accessing the precise model version described in the paper. In some cases, authors may prefer to put models on their own website, or to act as a point of contact for obtaining the code. Given the impermanence of websites and email addresses, this is not encouraged, and authors should consider improving the availability with a more permanent arrangement. Making code available through personal websites or via email contact to the authors is not sufficient. After the paper is accepted the model archive should be updated to include a link to the GMD paper.

Please provide the version number of PIC in the title of your revised manuscript.

As GitHub is not a persistent archive, and the code is not yet included in an official CESM version, please provide a persistent archive for the exact source code version used for the publication in this paper. As explained in https://www.geoscientific-model-development.net/about/manuscript_types.html the preferred reference to this release is through the use of a DOI which then can be cited in the paper. For projects in GitHub a DOI for a released code version can easily be created using Zenodo, see https://guides.github.com/activities/citable-code/ for details.

Finally note, that according to our new Editorial (v1.2) all data and analysis / plotting scripts should be made available.

Yours, Astrid Kerkweg

---

## Referee Comment (RC1) · Anonymous Referee #1 · 20 Jan 2021

- This paper addresses a "high bias in photosynthesis or gross primary productivity (GPP) at high latitudes" apparent in CLM5.0 model simulations." The paper is geared towards identifying issues in the current standard model version and making recommendations for modifications aimed at improving model performance for simulating carbon fluxes in the arctic-boreal zone (ABZ).

- The focus on accurately simulating seasonal C exchange in the high latitudes is important as it strongly controls the seasonal cycle of atmospheric CO2 - an aspect of Earth system model predictions that has been found to be inaccurately simulated and thus warrants closer attention and calls for improvement of available models. The present paper tackles this issue and thus promises to be an important contribution.

- I considered it particularly useful that the authors applied point-based simulations for

direct comparison with observed C fluxed at eddy covariance sites.

- However, several aspects of this study limit the usefulness of the presented research. I list major points below. Given that I have to raise these (in my view) rather fundamental issues, I cannot recommend this paper for publication in its current form.

- However, I was also appealed, e.g., by the useful focus and separating effects by phenology (start and end of season) and factors determining photosynthetic rates (Vcmax, Jmax). This could be explored further. Part of the challenge for the present study is that the apparent high bias in simulated GPP in the ABZ is the outcome of multiple potential factors that probably feed back on each other. E.g., high photosynthetic efficiency (light use efficiency) during the summer leads to high C assimilation which should enable an expansion of total leaf area which, in turn, should increase photosynthesis by increasing the fraction of absorbed light. The complication is that this is sort of a "chicken-or-egg problem" (What's the root cause?). A rigorous way forward to address this would be to disentangle contributions by, for example, prescribing seasonal leaf area from observations and calibrate parameters determining light use efficiency first. If the phenology routine was decoupled from other parts of the model (which it is not, see below), it could also be calibrated separately (without having to run the entire model). Then, once light use efficiency and phenology are well calibrated, one may calibrate parameters determining leaf area (e.g., allocation factors). In my view, this would be a promising way forward here. I understand however, that this may not be easily achievable. A "middle ground" could be found, e.g., if the model evaluation focused on these separate factors (phenology, light use efficiency, leaf area index) and tried to identify their relative contributions to model-observation mismatch in original and revised model versions. Having said that, I also consider that the model revision itself warrants reconsideration. I do not consider the model modifications to be recommendable for adoption for global simulations, as I argue below.

**Major**

- Modifications to make the model fit ABZ observations does not assure that the model performance is not deteriorated outside these biomes. This may seriously undermine the usefulness of proposed changes for global model simulations. Restricting the model applicability to the ABZ makes little sense for mitigating this limitation in view of common (global) applications of this model (e.g. Global Carbon Project simulations, CMIP, etc.). This limited scope of observations for informing model structure and calibrating parameters is all the more disappointing as authors note themselves that the model's current implementation, e.g., of the phenology routine or the temperature acclimation of Vcmax and Jmax, is based on data from a limited climatic range (essentially just the temperate zone). In this view, the manuscript seems to repeat a practice that has apparently been at the heart of poor model performance of the currently available CLM version. One way to resolve similar issues has been to assign PFT-specific parameters and thus accommodate for different parameter values to take effect in different biomes (this works in combination with achieving a realistic simulation of the PFT distribution). However, what is proposed here, e.g., for the phenology module, is to apply not just different parameters to a ABZ-typical PFT, but to change the *model structure* (Sect. 2.4.1). If I understood it correctly that authors propose to apply this structure only to PFTs growing in the ABZ, I have to raise concern about the implications of such PFT-specific parametrizations. This may seriously complicate interpretation global model predictions in future applications and the calibration of the model.

- Proposed modifications are very model-specific, don't make systematic use of available observational data of the affected variables, have little potential for adoption into other modelling frameworks, and have little potential to improve the general understanding of how simulations of C cycling in the ABZ can be improved (see major points below). In addition, by its focus on evaluating the (essentially global) model only with observations from the ABZ, the paper does not make clear whether the proposed model modifications improve global model performance metrics. For example, a test against global iLamb benchmarks would have been useful to demonstrate the

usefulness of proposed changes. Let me clarify my concerns about the proposed modifications:

- Spring phenology: The proposed modification relies on internally simulated quantities as arguments to the phenology function (soil temperature, snow depth). This implies the risk of undesired effects on simulated phenology caused by modifications (possibly in the future) to the snow or soil temperature routines. Such "feedbacks" between different parts of the model complicate model development and the identification of root causes for model bias. The chosen formulation of spring phenology is all the more surprising since this complication is avoided by the use of growing-degree-day-based models that are standard and well-established (see e.g., Richardson et al., 2018; Hufkens et al.., 2018) for robust simulations of spring phenology (with some modifications like chilling requirement).

- Temperature acclimation of Vcmax and Jmax: Authors suggest to revert the formulation of temperature-acclimation from the currently implemented version designed following Kattge & Knorr (2007) to a previous version based on Leuning (2002). This happens to improve model performance in CLM5.0 and is justified here by reference to the limited representativity of the parametrisation proposed by Kattge & Knorr (2007). The manuscript does not clarify the structure of the parametrisations of the two versions. Either way, this change is hardly justifiable by improved process understanding. Authors also refer to Kumarathunge et al. (2019) who recently updated the analysis of Kattge & Knorr (2007) using a much extended dataset, now encompassing data from a wider climatic range. It remains elusive why the parametrisations proposed by Kumarathunge et al. (2019) were not used here. This would have been a potentially useful modification of the CLM model, based on improved understanding and a wider and more robust observational basis. [references given in the manuscript of Birch et al.]

- The modified initialisation of Vcmax and Jmax at the start of the season (Sect. 2.4.5) is specific to a particular module (LUNA) within CLM5.0 and thus has little relevance

for adoption into modelling frameworks outside or for informing process understanding. In my view, this rather seems like a bugfix than a model improvement worth publication outside a CLM-specific technical report.

- Carbon allocation: Alternative choices (static allocation with different root:leaf allocation ratios, dynamic allocation, Sect. 2.4.6) were tested. However, as I understand it, the tests appear to be evaluated with respect to model performance in simulating GPP. Authors limit the justification for selecting a particular value by reference to a small number of references. This approach to model development makes no systematic use of relevant observational data on allocation patterns itself nor of calibration methods, and runs the risk of being affected by compensating errors between model performance in simulating allocation and GPP (authors do not demonstrate that the chosen modification of the allocation parametrisation actually improves simulated allocated patterns).

**References**

Hufkens K., Basler J. D., Milliman T. Melaas E., Richardson A.D. 2018Ăă[An integrated phenology modelling framework in R: Phenology modelling with phenor. Methods in Ecology & Evolution](http://onlinelibrary.wiley.com/doi/10.1111/2041-210X.12970/full), 9: 1-10.

Richardson, A.D., Hufkens, K., Milliman, T., Aubrecht, D.M., Chen, M., Gray, J.M., Johnston, M.R., Keenan, T.F., Klosterman, S.T., Kosmala, M., Melaas, E.K., Friedl, M.A., Frolking, S. 2018.Ăă[Tracking vegetation phenology across diverse North American biomes using PhenoCam imagery](https://www.nature.com/articles/sdata201828). Scientific Data, 5, 180028.

---

## Referee Comment (RC2) · Anonymous Referee #2 · 21 Jan 2021

This manuscript aims to correct the biases in representing circumpolar carbon cycling by the CLM5.0 model. The authors did a good job pinpointing model deficiencies in CLM5.0 responsible for these biases. To address these deficiencies, they focused on point-based simulations and compared directly with PFT-specific seasonal cycle of carbon fluxes in the model development. The paper is well-written and easy to follow, and their model recommendation show significant improvement in its capacity in capturing the mean seasonal characteristics of Arctic-Boreal carbon cycling.

As many terrestrial biosphere models show poor performance in simulating the seasonal cycle of CO2 exchange, especially in the high latitudes, this paper could potentially offer important insights to the wider modeling and scientific community. The authors faced a challenging task as observations are sparse in the ABZ, and they im-

plemented the model improvements following clear and logical steps. However, there are two major issues limiting the broader appeal (i.e., contributing to a future stock version of CLM or motivating changes in other models) of this research.

Firstly, although mean seasonal carbon cycle is important, other features such as the magnitude change of carbon fluxes, their interannual variability, and the mean carbon sink, are also essential. It would reassure the readers if the authors could demonstrate in more detail that other aspects of the model do not become worse after the changes. Currently the authors rely heavily on FLUXCOM data in validations, how reliable is the PFT-specific output of FLUXCOM? Could other observation-based datasets, such as atmospheric inversions, and observed phenology data be used in the comparison as well in the benchmarking? If any of the proposed changes are not limited to ABZ, it would make sense to also show results for a global-scale benchmarking. Otherwise, it would be helpful to quantify the contribution of updated ABZ carbon fluxes to global carbon fluxes.

Secondly, it was not always clear how the authors arrive at their proposed changes. For phenology onset, was there any attempt to improve the scheme based on growing degree days? Were any quantitative criteria (in addition to visual inspection of Fig. S2) used in determining the environmental metrics used? For temperature acclimation, why was Kattge and Knorr used over Leuning in previous CLM development? Was there any sensitivity testing in the updated parameters (i.e., carbon allocation)? As only four sites were used in model development, and that the mechanisms changed show compensating effects in changing the seasonal carbon cycle, the new model parameters might be poorly constrained and could introduce new biases in the model.

Minor points: - Figure 3, the label "Temp. Scaling"is potentially misleading as it also include daylight scaling, and some of the line colors are hard to differentiate. - Why did the model still perform poorly in Figure 4 a) and c) despite the changes? - The discussion could be more succinct and repeat less information from previous sections. It would interest potential readers if the authors could discuss if other TBMs also have

similar deficiencies as identified in CLM5.0.

Interactive
comment

---

## Referee Comment (RC3) · Anonymous Referee #3 · 8 Feb 2021

Birch et al improve the representation of Arctic-boreal zone CO2 fluxes in CLM5 through examination of model biases compared to gridded flux products and eddy co-variance tower data. They implement process-level changes that improve the uptake phenology and extent of productivity for specific plant functional types. Overall, the paper is well written, detailed, and comprehensive of this highly relevant topic and worthy of publication in GMD with minor revisions.

The entire model evaluation and analysis is based on the assumption that the uptake/respiration component flux partitioning in the FLUXCOM and at EC sites is correct. However, there is no discussion of the uncertainties inherent in these products. GPP cannot be independently observed and thus all values are simulated. Certainly, we can rely on these partitioning products for understanding, but an expanded discussion of

this topic should be included in the methods section along with the product descriptions to boost confidence in the analysis.

Certain portions of the results and discussion, such as lines 372-376, are redundant with methodology described in section 2.4. I encourage the authors to review connections between these sections to remove unneeded words and references to previously described processes and/or results.

Additional minor comments:

Line 56. If satellite products are unavailable in the winter, then they are not complete in time. Perhaps be more specific about to which kinds of products you are referring.

Line 106: Less biased compared to which other product?

Line 151: What impact does the choice of a site as development vs validation make on the results? If you swap sites between categories would you get a different answer?

Line 152: "evaluation" is preferred over "validation"

Line 187: How is "GPP onset" defined? GPP > 0? Or when NEE begins to decrease?

Figure 1: Inconsistent capitalization of FLUXCOM

Section 3.1: NEE changes between model versions are fairly small. As with overview comment above, how can you know that large changes to component fluxes are needed when there is little change to the NEE (which is what is actually observed).

Figure 3: Add units to second row of plots, PFT abbreviations should be defined in the legend Line 346: Missing a word, "note"?

Line 566: Why does separating the sites lead to mitigating the lack of data? Are you not further decreasing the amount of data your changes are based on?

Live 590: Global carbon budget change seems like it could be calculated only based on the changes made for the ABZ. This would increase the value of the study and

highlight the global importance of understanding the ABZ CO2 fluxes.

---

## Author Comment (AC1) · 5 Apr 2021

Dear Dr. Sato, GMD Editors, and Referees, Please see the attached supplement for all our responses to your helpful comments. Thank you for the time you took to review our work and we look forward to continuing this conversation.

Please also note the supplement to this comment: https://gmd.copernicus.org/preprints/gmd-2020-365/gmd-2020-365-AC1-supplement.pdf

---

## Author Response (AR1)

Woodwell Climate Research Center
149 Woods Hole Rd, Falmouth, MA 02540
lbirch@woodwellclimate.org, brogers@woodwellclimate.org
Leah Birch and Brendan Rogers

Dear Dr. Sato, GMD Editors, and Reviewers,

Thank you for your helpful comments throughout this process. We have revised our manuscript and hopefully addressed your questions, both here and in the main paper. Line numbers referenced here relate to the revised manuscript. We have included a diff'ed manuscript after our point by point responses here. We believe that your critiques helped us bring clarity to our manuscript and reduce redundancy, particularly in our methods and discussion sections.

Thank you again for the time that you have taken to review our work.

Sincerely yours,

Leah Birch and Brendan Rogers

On behalf of all contributing authors

**Revisions**

**Response to Referee 1**

- - This paper addresses a "high bias in photosynthesis or gross primary productivity (GPP) at high latitudes" apparent in CLM5.0 model simulations." The paper is geared towards identifying issues in the current standard model version and making recommendations for modifications aimed at improving model performance for simulating carbon fluxes in the arctic-boreal zone (ABZ).

  - The focus on accurately simulating seasonal C exchange in the high latitudes is important as it strongly controls the seasonal cycle of atmospheric CO2 - an aspect of Earth system model predictions that has been found to be inaccurately simulated and thus warrants closer attention and calls for improvement of available models. The present paper tackles this issue and thus promises to be an important contribution.

  - I considered it particularly useful that the authors applied point-based simulations for direct comparison with observed C fluxed at eddy covariance sites.

  - However, several aspects of this study limit the usefulness of the presented research. I list major points below. Given that I have to raise these (in my view) rather fundamental issues, I cannot recommend this paper for publication in its current form.

  Thank you for taking the time to read through our paper. We also believe that carbon cycling in the ABZ is an important component of ESMs that requires analysis and development to address current biases. We hope to clarify our methods and alleviate some of your concerns. We also believe this study can be a stepping stone to help other researchers in further diagnosing issues in CLM and ultimately improve our predictive capabilities in Earth system modeling.

- - However, I was also appealed, e.g., by the useful focus and separating effects by phenology (start and end of season) and factors determining photosynthetic rates (Vcmax, Jmax). This could be explored further. Part of the challenge for the present study is that the apparent high bias in simulated GPP in the ABZ is the outcome of multiple potential factors that probably feed back on each other. E.g., high photosynthetic efficiency (light use efficiency) during the summer leads to high C assimilation which should enable an expansion of total leaf area which, in turn, should increase photosynthesis by increasing the fraction of absorbed light. The complication is that this is sort of a "chicken-oregg problem" (What?s the root cause?). A rigorous way forward to address this would be to disentangle contributions by, for example, prescribing seasonal leaf area from observations and calibrate parameters determining light use efficiency first. If the phenology routine was decoupled from other parts of the model (which it is not, see below), it could also be calibrated separately (without having to run the entire model). Then, once light use efficiency and phenology are well calibrated, one may calibrate parameters determining leaf area (e.g., allocation factors). In my view, this would be a promising way forward here. I understand however, that this may not be easily achievable. A "middle ground" could be found, e.g., if the model evaluation focused on these separate factors (phenology, light use efficiency, leaf area index) and tried to identify their relative contributions to model-observation mismatch in original and revised model versions.

We understand the appeal of an idealized approach that completely separates model components, and agree on its utility. However, these model components, such as light use efficiency and leaf area mentioned by the reviewer, are not orthogonal and indeed are highly inter-dependent in the model. Additionally, because of these interdependencies, the contribution of one parameter or model formulation to overall bias depends on the other parameters or model formulations, making this type of analysis extremely challenging. We did, however, vary parameters independently to understand their contributions to model biases within point simulations at flux sites, and we have included a few more point simulation figures in the Supplement S12-15. In presenting our results, we found it consistently more straightforward and compelling to show changes sequentially and in an additive manner.

- - Having said that, I also consider that the model revision itself warrants reconsideration. I do not consider the model modifications to be recommendable for adoption for global simulations, as I argue below.

  Modifications to make the model fit ABZ observations does not assure that the model performance is not deteriorated outside these biomes. This may seriously undermine the usefulness of proposed changes for global model simulations. Restricting the model applicability to the ABZ makes little sense for mitigating this limitation in view of common (global) applications of this model (e.g. Global Carbon Project simulations, CMIP, etc.). This limited scope of observations for informing model structure and calibrating parameters is all the more disappointing as authors note themselves that the model?s current implementation, e.g., of the phenology routine or the temperature acclimation

of Vcmax and Jmax, is based on data from a limited climatic range(essentially just the temperate zone). In this view, the manuscript seems to repeat a practice that has apparently been at the heart of poor model performance of the currently available CLM version. One way to resolve similar issues has been to assign PFT-specific parameters and thus accommodate for different parameter values to take effect in different biomes (this works in combination with achieving a realistic simulation of the PFT distribution). However, what is proposed here, e.g., for the phenology module, is to apply not just different parameters to a ABZ-typical PFT, but to change the *model structure* (Sect. 2.4.1). If I understood it correctly that authors propose to apply this structure only to PFTs growing in the ABZ, I have to raise concern about the implications of such PFT-specific parametrizations. This may seriously complicate interpretation global model predictions in future applications and the calibration of the model.

We wholeheartedly agree that beneficial changes in one region of the model should not result in skill deterioration for the other regions, particularly for a model such as CLM that gets used primarily for global assessments. Our manuscript is focused on the ABZ, which is why we were running regional CLM simulations. We agree that not all model development choices presented here would improve carbon cycling outside of the ABZ. However, most of our model changes are ABZ PFT specific, meaning they would not directly affect any regions outside of the ABZ. To quantify the impact of our model development on global performance, we ran a global simulation as described in Section 3.4 at Line 460, which we moved from the Supplement to the main part of the paper. Because of the improvements at high latitudes, our changes for the ABZ had the effect of improving CLM's representation of global GPP.

The use of PFTs as mentioned by the referee is an effective way to accommodate biome-specific processes within global models. PFT phenology schemes are already implemented in CLM and have been used for many years (i.e. evergreen phenology, deciduous phenology, and stress deciduous phenology). To address biases when extrapolating phenology schemes from temperate to high latitudes, we added an ABZ deciduous phenology scheme, which we describe and clarify in Section 2.4.1 on lines 185-204.

- - Proposed modifications are very model-specific, don?t make systematic use of available observational data of the affected variables, have little potential for adoption into other modelling frameworks, and have little potential to improve the general understanding of how simulations of C cycling in the ABZ can be

improved (see major points below). In addition, by its focus on evaluating the (essentially global) model only with observations from the ABZ, the paper does not make clear whether the proposed model modifications improve global model performance metrics. For example, a test against global iLamb benchmarks would have been useful to demonstrate the usefulness of proposed changes. Let me clarify my concerns about the proposed modifications:

We focused our simulations on the ABZ and all but two of our model changes only impact that region. As mentioned, we only advocate for all of our model developments in the ABZ. That said, we did test a global simulation with all of our model changes and assessed performance using ILAMB (Figure 6). We found general and significant improvements in the global simulation and have moved this result from the supplement (where it was previously) to the main part of the paper. Though our ABZ-focused development improved the global simulation, anyone seeking to use this model set up should be aware of the change from Kattge and Knorr to Leuning, which will be discussed below in response to a following question.

- - Spring phenology: The proposed modification relies on internally simulated quantities as arguments to the phenology function (soil temperature, snow depth). This implies the risk of undesired effects on simulated phenology caused by modifications (possibly in the future) to the snow or soil temperature routines. Such "feedbacks" between different parts of the model complicate model development and the identification of root causes for model bias. The chosen formulation of spring phenology is all the more surprising since this complication is avoided by the use of growing-degreeday-based models that are standard and well-established (see e.g., Richardson et al., 2018; Hufkens et al.., 2018) for robust simulations of spring phenology (with some modifications like chilling requirement).

We appreciate the reviewer?s concern. Here we note that such internal feedbacks are ubiquitous in complex land surface models such as CLM, yet this should not preclude the development of physically-based parameters for processes such as spring onset. Fundamentally, what determines spring onset in the ABZ is related to the freeze/thaw state of the ground as this impacts the availability of soil moisture and root metabolic activity as described in our Methods section, lines 195-200. We agree that phenology approaches based on growing degree days are useful. However, the vast majority of observations informing these formulations are from latitudes lower than the ABZ with most validation studies also focused on lower latitude regions. The studies that do

include ABZ phenology depicted late onset timing, which was the bias that we intended to improve. In the case of CLM, the GDD threshold was parameterized based on the temperate United States for three years in the 1990s. We expanded on our rationale for moving away from the GDD approach in the discussion, lines 472-488.

- - Temperature acclimation of Vcmax and Jmax: Authors suggest to revert the formulation of temperature-acclimation from the currently implemented version designed following Kattge Knorr (2007) to a previous version based on Leuning (2002). This happens to improve model performance in CLM5.0 and is justified here by reference to the limited representativity of the parametrisation proposed by Kattge Knorr (2007). The manuscript does not clarify the structure of the parametrisations of the two versions. Either way, this change is hardly justifiable by improved process understanding.

  Kattge and Knorr did improve process understanding, but only in temperate regions. The structures of the parameterizations are largely similar between the two methods, with the exception that Kattge and Knorr introduced an additional term for acclimation between 11 and 35C, as described in the Methods section. This unfortunately introduces more problems than it solves in the ABZ. We present an expanded discussion on this in the Methods (245-264) and Discussion (line 506-517) sections.

- Authors also refer to Kumarathunge et al. (2019) who recently updated the analysis of Kattge Knorr (2007) using a much extended dataset, now encompassing data from a wider climatic range. It remains elusive why the parametrisations proposed by Kumarathunge et al. (2019) were not used here. This would have been a potentially useful modification of the CLM model, based on improved understanding and a wider and more robust observational basis. [references given in the manuscript of Birch et al.]

  The Kumarathunge parameterizations are a promising step forward utilizing a wider temperature range. We hope that the modeling community tests the implications of adopting them for temperate regions, as they improved variance in prediction relative to Kattgne and Knorr. However, Kumarathunge would significantly change the current model structure in CLM dramatically, which we found to be outside of the scope of this paper. Moreover, they only included Utqiagvik (Barrow), Alaska and a site in Finland as part of the ABZ, so the scheme can still be considered an extrapolation at high latitudes, as described on lines 515-517.

- - The modified initialisation of Vcmax and Jmax at the start of the season (Sect. 2.4.5) is specific to a particular module (LUNA) within CLM5.0 and thus has little relevance for adoption into modelling frameworks outside or for informing process understanding. In my view, this rather seems like a bugfix than a model improvement worth publication outside a CLM-specific technical report.

  We agree that whether or not this was a development or bug fix could be debated. Nevertheless, we found the issue to be very influential in causing seasonal biases. Given CLM's wide spread use for regional and global applications, assessment reports, and incorporation in other models such as WRF, we believe it is important knowledge to disseminate. To address the reviewer's critique, we have moved some of this section to the Supplement because we do think the equations are better suited there. We have also added a plot to the Supplement (Fig. S15), which depicts the sensitivity of GPP to the initial constant values of Equations 2/3.

- - Carbon allocation: Alternative choices (static allocation with different root:leaf allocation ratios, dynamic allocation, Sect. 2.4.6) were tested. However, as I understand it, the tests appear to be evaluated with respect to model performance in simulating GPP.

  The reviewer is correct in that we evaluated the effects of allocation parameters on GPP most closely, but we also evaluated the effects on NEE and TER.

- Authors limit the justification for selecting a particular value by reference to a small number of references.

  This is an area where more observations would certainly useful. However, there is a disconnect between real world allocation parameters and allocation in CLM. The CLM PFT categories lump multiple plant species, across wet and dry biomes, together into broad categories. Since allocation changes across plant species, matching observations exactly is not realistic. We did test a small range of different combinations based on observations and placed an example in the Supplement (Fig. S14). Ultimately, we settled on attempting to create the relative differences in allocation for different PFTs, whereas before allocation was the same across all PFTs as described in our Methods Section line 283.

- This approach to model development makes no systematic use of relevant observational data on allocation patterns itself nor of calibration methods, and runs the risk of being affected by compensating errors between model performance

in simulating allocation and GPP (authors do not demonstrate that the chosen modification of the allocation parametrisation actually improves simulated allocated patterns).

Allocation patterns in CLM are very dependent on the input parameters, such that the resulting patterns followed the input parameters closely. Thus, the carbon fluxes are one of the best proxies we have for evaluating model performance as it is affected by allocation. We are currently not aware of mapped allocation values, but that would be a very valuable dataset to modeling efforts.

**Response to Referee 2**

- This manuscript aims to correct the biases in representing circumpolar carbon cycling by the CLM5.0 model. The authors did a good job pinpointing model deficiencies in CLM5.0 responsible for these biases. To address these deficiencies, they focused on point-based simulations and compared directly with PFT-specific seasonal cycle of carbon fluxes in the model development. The paper is well-written and easy to follow, and their model recommendation show significant improvement in its capacity in capturing the mean seasonal characteristics of Arctic-Boreal carbon cycling.

As many terrestrial biosphere models show poor performance in simulating the seasonal cycle of CO2 exchange, especially in the high latitudes, this paper could potentially offer important insights to the wider modeling and scientific community. The authors faced a challenging task as observations are sparse in the ABZ, and they implemented the model improvements following clear and logical steps. However, there are two major issues limiting the broader appeal (i.e., contributing to a future stock version of CLM or motivating changes in other models) of this research.

We thank the reviewer for appreciating the importance of this work. We localized most of our model improvements to the ABZ, so incorporation into CLM is possible and ongoing with many of our changes. We believe that keeping NCAR informed throughout the model development has been key to seeing our model improvements actualized in the main branch.

- Firstly, although mean seasonal carbon cycle is important, other features such as the magnitude change of carbon fluxes, their interannual variability, and the mean carbon sink, are also essential. It would reassure the readers if the

authors could demonstrate in more detail that other aspects of the model do not become worse after the changes.

The reviewer raises a good point in that we focused heavily on one feature for development. Fortunately, the ILAMB framework allows us to investigate the effects on other critical variables and interannual variability. We find general improvement across carbon fluxes, heat fluxes, and moisture fluxes, lines 449-465. Thus, extrapolating temperate schemes for vegetation processes to the ABZ can impact more than just carbon fluxes, increasing biases.

- Currently the authors rely heavily on FLUXCOM data in validations, how reliable is the PFT-specific output of FLUXCOM? Could other observation-based datasets, such as atmospheric inversions, and observed phenology data be used in the comparison as well in the benchmarking?

The PFT-specific FLUXCOM product uses the same methods as FLUXCOM, and we have added a better description of FLUXCOM to our Methods Section. We chose to use ILAMB for additional validation of our model improvements, which provides a comprehensive assessment of key modeled variables. Indeed, there are additional data sets that could be leveraged. However, each comes with its own sets of limitations. For example, gridded phenology products tend to be produced using optical imagery, which often does not correspond well with $CO_2$ fluxes in shoulder seasons (line 480). Atmospheric inversions often disagree on the magnitude and trend in net $CO_2$ fluxes at high latitudes, in part due to coarse grid resolutions (Gourdji, 2012; Schuh, 2013). We therefore believe that a focus on the widely-used and well-validated data sets in FLUXCOM and ILAMB provides an adequate basis for model validation.

References:

Gourdji, Sharon M., et al. "North American CO 2 exchange: inter-comparison of modeled estimates with results from a fine-scale atmospheric inversion." Biogeosciences 9.1 (2012): 457-475.

Schuh, Andrew E., et al. "Evaluating atmospheric CO2 inversions at multiple scales over a highly inventoried agricultural landscape." Global change biology 19.5 (2013): 1424-1439.

- If any of the proposed changes are not limited to ABZ, it would make sense to also show results for a global-scale benchmarking. Otherwise, it would be helpful to quantify the contribution of updated ABZ carbon fluxes to global carbon fluxes.

We agree with the reviewer on the importance of including a global simulation to test the impact of our changes for the ABZ on global model performance. To highlight this, and the global improvement offered by better carbon cycling at high latitudes, we have moved our test of a global simulation from the supplement to the main part of the paper (lines 460-465; Figure 6). We want to caution on the extrapolation of an ABZ-specific improvement catalogue to the full globe, and please also refer to our response to Reviewer 1 on this topic.

- Secondly, it was not always clear how the authors arrive at their proposed changes. For phenology onset, was there any attempt to improve the scheme based on growing degree days?

We appreciate the reviewer?s comment and included an expanded discussion on this topic in the manuscript (lines 472-488). We did investigated other several growing degree day (GDD) schemes. However, as noted above in response to Reviewer 1, we did not identify one that we was both well- validated for the ABZ and performed well there within CLM. In essence, we would have been trading one extrapolated scheme for another, which is why we decided to test a physically-based threshold approach.

- Were any quantitative criteria (in addition to visual inspection of Fig. S2) used in determining the environmental metrics used?

We tested multiple threshold values, but ultimately did rely on differences being within yearly standard deviation values to guide our decision. We found those plots messier to include than Fig. S2, which is why we made that decision on that figure to help depict part of the process.

- For temperature acclimation, why was Kattge and Knorr used over Leuning in previous CLM development?

Kattge and Knorr were proposed as an addition to CLM to add better process understanding. Leuning did not allow for acclimation of C3 photosynthesis and leaf respiration to changing $CO_2$ levels. However, Kattge and Knorr have temperature limitations and no ABZ vegetation included in their study. We believe that process understanding is important, but in this case, it led to more biases in the ABZ. It may be most useful to allow the user to choose the scheme that they want in CLM5.0, but for regional ABZ simulations, we recommend Leuning.

- Was there any sensitivity testing in the updated parameters (i.e., carbon allocation)?

Yes, we did test a range of values around the parameters that we ended up choosing. Some of the specific intermediate flux tower results are in the Supplement Figures 4, 8-9 and 12-14.

- As only four sites were used in model development, and that the mechanisms changed show compensating effects in changing the seasonal carbon cycle, the new model parameters might be poorly constrained and could introduce new biases in the model.

We attempted to avoid over fitting the model by splitting the data in test (n = 4) and evaluation (n = 6) data for the flux towers. Notably, our development performed consistently well at evaluation sites (Fig. 4). We also found the ILAMB results to be highly favorable, including independent observation data on carbon fluxes and energy balance. Nevertheless, we cannot definitively show that we did not introduce, new biases that did not appear within our evaluation framework. We believe that we have addressed a number of biases at high latitudes as we have systemically removed model logic from the ABZ that was based on extrapolation from the temperate zone. We look forward to seeing more improvements of ABZ simulations.

- Minor points: - Figure 3, the label ?Temp. Scaling?is potentially misleading as it also include daylight scaling, and some of the line colors are hard to differentiate.

Fair point, we renamed this as Param. Scaling and thickened the lines in the plot.

- - Why did the model still perform poorly in Figure 4 a) and c) despite the changes?

We would actually argue that 4a is a much better simulation than the originally inactive, effectively dead vegetation that occurred there previously. Offset is early compared to the tower, but the simulated GPP in December is likely a partitioning/gap filling error. In regards to 4c, we decreased maximum GPP by almost half, but it is still high and an area that we noted needs work in CLM, which is part of what we are hoping to draw attention to this paper. There are many areas for future work in improving ABZ simulations.

- - The discussion could be more succinct and repeat less information from previous sections.

We agree and have attempted to rectify this by focusing more on limitations and avenues for future research throughout the Discussion.

- It would interest potential readers if the authors could discuss if other TBMs also have similar deficiencies as identified in CLM5.0.

  Given the widespread use of CLM and it's incorporation in other models such as WRF, we believe that documenting these biases and possible solutions should still have broad appeal. Our main conclusion is that the extrapolation of temperate schemes to the ABZ is the root of many biases, which also generally applies to other models. Additionally, wherever possible, we have noted similarities to other models (Lines).

**Response to Referee 3**

- Birch et al improve the representation of Arctic-boreal zone CO2 fluxes in CLM5 through examination of model biases compared to gridded flux products and eddy covariance tower data. They implement process-level changes that improve the uptake phenology and extent of productivity for specific plant functional types. Overall, the paper is well written, detailed, and comprehensive of this highly relevant topic and worthy of publication in GMD with minor revisions.

  We appreciate the reviewer?s kind words and acknowledgement of the importance of our work.

- The entire model evaluation and analysis is based on the assumption that the uptake/respiration component flux partitioning in the FLUXCOM and at EC sites is correct. However, there is no discussion of the uncertainties inherent in these products. GPP cannot be independently observed and thus all values are simulated. Certainly, we can rely on these partitioning products for understanding, but an expanded discussion of this topic should be included in the methods section along with the product descriptions to boost confidence in the analysis.

  Excellent point, we have added more details to the Methods section (lines 130-141). This is mostly to clarify the process of PFT specific FLUXCOM and boost confidence in this invaluable dataset for carbon fluxes.

- Certain portions of the results and discussion, such as lines 372-376, are redundant with methodology described in section 2.4. I encourage the authors to review connections between these sections to remove unneeded words and references to previously described processes and/or results.

Thanks for this comment. We have attempted to reduce redundancy throughout the paper, and removed these lines in particular.

- Additional minor comments: Line 56. If satellite products are unavailable in the winter, then they are not complete in time. Perhaps be more specific about to which kinds of products you are referring.

We appreciate this comment and have attempted to make this clear in the text (lines 56-60). Here we are basically just saying noting that satellite measurements even in winter are challenging, which necessitates the use of models such as CLM.

- Line 106: Less biased compared to which other product?

GSWP was found to be the least biased compared to other reanalysis products used with CLM. CRUNCEP is the most well known reanalysis dataset used in CLM5.0 but Lawrence 2019 mentioned there used a few reanalysis products (line 106).

- Line 151: What impact does the choice of a site as development vs validation make on the results? If you swap sites between categories would you get a different answer?

If we were doing a more formal model calibration study, we believe it would have been possible to find slightly different parameter results between categories. However, our parameter tests were largely pretty coarse, which led to pretty clear parameter choices if we were hoping to have modeled GPP within the standard deviation limits of observational products. Given the uncertainty of the available data products, including those we used (discussed in this response letter and now more comprehensively in the manuscript), we chose to avoid over-fitting by opting for physically-based model development that resulted in seasonal CO2 fluxes being simulated within the bounds of observational uncertainty, both for our training flux sites and our withheld evaluation sites (Fig. 4). Swapping sites between the categories of development and validation may have resulted in slightly different development choices, but our improvements clearly improve performance across all sites, and our fundamental conclusions about avoiding model formulations based on data from temperate biomes for the ABZ would remain robust

- Line 152: ?evaluation? is preferred over ?validation?

We agree, better word choice.

- Line 187: How is ?GPP onset? defined? GPP ¿ 0? Or when NEE begins to decrease?

  We define GPP onset as $GPP > 0$. However, in CLM, NEE begins to decrease at the same time, so it was not a choice we needed to make. If we were examining daily GPP, then this distinction may have mattered more.

- Figure 1: Inconsistent capitalization of FLUXCOM

  We have attempted to fix this throughout. Thanks for pointing it out.

- Section 3.1: NEE changes between model versions are fairly small. As with overview comment above, how can you know that large changes to component fluxes are needed when there is little change to the NEE (which is what is actually observed).

  This is a good point. NEE is the quantity actually measured, but GPP and respiration are the processes directly simulated in CLM. We therefore believe it is more direct to focus on GPP and trust the partitioning methods. The final NEE simulation is ultimately improved according to ILAMB (line 455).

- Figure 3: Add units to second row of plots, PFT abbreviations should be defined in the legend

  We added the units, and added the PFT abbreviations to figure captions instead, which we think is easier to read.

- Line 346: Missing a word, ?note??

  Yes, thank you for finding that.

- Line 566: Why does separating the sites lead to mitigating the lack of data? Are younot further decreasing the amount of data your changes are based on?

  We apologize for this sentence. We meant to say that we attempted to prevent over fitting by splitting the data into calibration and evaluation data. With the few ABZ sites available, we did not want to tune the model exactly to a dozen sites. Our development process involved iteratively testing possible parameters at four specific PFT sites. We then evaluated our final choices at the sites withheld before running a regional CLM simulation, which at least partially mitigated the lack of field observations.

- Live 590: Global carbon budget change seems like it could be calculated only based on the changes made for the ABZ. This would increase the value of

This is a good point. We have attempted to highlight more global results by bringing Figure 6 from the Supplement to the main paper. We also state changes in NEE more clearly, lines 440-445.

**Response to Interactive Comment**

Regarding the comments from Astrid Kerkweg on data availability, we have left the github reference for now we believe the discussion there is also informative to anyone interested in using the model and knowing it's current state with NCAR. We will be ultimately be archiving it with Zenodo as suggested. We find archiving large amounts of climate model simulations a difficult prospect unfortunately, but we will make our data available to anyone who asks.